# Partial identification of the maximum mean discrepancy with mismeasured data

**Ron Nafshi**[1]                    **Maggie Makar**[2]

[1]Fintica AI, Tel Aviv, Israel
[2]University of Michigan, Ann Arbor, MI

## Abstract

Nonparametric estimates of the distance between two distributions such as the Maximum Mean Discrepancy (MMD) are often used in machine learning applications. However, the majority of existing literature assumes that error-free samples from the two distributions of interest are available. We relax this assumption and study the estimation of the MMD under $\epsilon$-contamination, where a possibly non-random $\epsilon$ proportion of one distribution is erroneously grouped with the other. We show that under $\epsilon$-contamination, the typical estimate of the MMD is unreliable. Instead, we study partial identification of the MMD, and characterize sharp upper and lower bounds that contain the true, unknown MMD. We propose a method to estimate these bounds, and show that it gives estimates that converge to the sharpest possible bounds on the MMD as sample size increases, with a convergence rate that is faster than alternative approaches. Using three datasets, we empirically validate that our approach is superior to the alternatives: it gives tight bounds with a low false coverage rate.

## 1 INTRODUCTION

Many machine learning methods rely on comparing distances between distributions, with applications ranging from single cell sequencing [Schiebinger et al., 2019] to causal inference [Johansson et al., 2016]. The Maximum Mean Discrepancy (MMD) [Gretton et al., 2012] has emerged as a particularly useful nonparametric notion of distance between distributions. It has been widely used in robust predictive and reinforcement learning [Kumar et al., 2019, Makar et al., 2022, Li et al., 2017, Oneto et al., 2020, Veitch et al., 2021, Goldstein et al., 2022], fairness applications [Prost et al., 2019, Madras et al., 2018, Makar and D'Amour, 2022, Louizos et al., 2015] and distributionally robust optimization [Staib and Jegelka, 2019, Kirschner et al., 2020] among others. Despite its importance and widespread use, the majority of existing work using the MMD assumes that observed samples are measured without error. As we show in this work, if this assumption does not hold, the typical MMD estimate is unreliable.

Here, we study the estimation of the MMD where one of the samples observed is measured with error. Specifically, we consider $\epsilon$-contamination, where a possibly non-random $\epsilon$ proportion of one of the two variables is erroneously grouped with the other. This mismeasurement mechanism arises in many important applications. One example of this setting arises from the fairness literature. For example, in settings where we wish to assess if a model gives different predictions across different race groups. Here $\epsilon$-contamination arises if some non-random $\epsilon$ proportion of one race group is incorrectly grouped with the other. Beyond fairness, $\epsilon$-contamination arises – for example – when trying to identify if there are biomarkers for Myocardial Infarction (MI). In this setting, we can use the MMD to detect differences in genome sequences between healthy individuals and patients with myocardial MI. Detecting differences between the two groups is complicated due to undiagnosed "silent" MI cases. These silent MI cases represent $\epsilon$-contamination that occurs non-randomly: women's MI cases are more likely to go undiagnosed compared to men [Merz, 2011].

In this paper, we show that the typical MMD estimates are unreliable when the data is collected with $\epsilon$-contamination. Instead, we resort to a partial identification approach, where we estimate upper and lower bounds on the MMD. We characterize upper and lower bounds that are credible, meaning that they contain the true unknown MMD, and sharp, meaning they cannot be made tighter without additional assumptions. Importantly, these bounds are identifiable using the observed contaminated data and an estimate of $\epsilon$. We develop an estimation approach to compute the upper and lower bounds and analyze its behavior in finite samples. Our analysis shows that our approach gives estimates that con-

verge to the sharpest possible upper and lower bounds as the sample size increases at a rate faster than the alternatives.

**Our contributions are summarized as follows**: **(1)** We show that under $\epsilon$-contamination the typical estimates of the MMD are unreliable, **(2)** We characterize sharp upper and lower bounds on the unknown MMD that are identifiable using only the observed contaminated data, and an estimate of $\epsilon$, **(3)** We propose an estimation approach to compute the upper and lower bounds and analyze its behavior in finite samples showing that its convergence to the true upper and lower bounds depends on the sample size and the value of $\epsilon$, **(4)** We apply our approach to 3 datasets showing that it achieves a superior performance compared to alternative approaches, **(5)** We analyze the sensitivity of our approach to incorrect values of $\epsilon$ and give practical guidance on what to do if the true value of $\epsilon$ is unknown.

**Related work.** Most existing work on the MMD focuses on establishing statistically and computationally efficient estimators of the difference between two distributions under the assumption that the observed samples are error-free [Gretton et al., 2012, 2009, Schrab et al., Domingo-Enrich et al., 2023]. However, to our knowledge, the only existing work that tackles the challenge of measurement error is in the context of survival analysis, where the measurement error model arises from the classical right-censoring of the data [Fernández and Rivera, 2021]. By contrast, we study a different measurement error mechanism and suggest methods for partial identification of the MMD.

In the fairness literature, where comparisons between outcomes of different groups is important, Kallus et al. [2022] consider measurement error in the sensitive attribute. They consider a setting where we only have access to an imperfect proxy of the protected class membership and show that typical fairness metrics such as demographic parity and equalized odds are not identifiable. Similar to our work, they develop methods for partial identification of these metrics. A key difference between Kallus et al. [2022] and our work is that the former focuses on comparing a single moment (the mean) of two distributions whereas our work allows a more rigorous comparison of infinitely many moments of two distributions. We also stress that while the methods presented here could be used in a fairness context, they are more widely applicable to any setting where we wish to compare two distributions.

## 2 PRELIMINARIES

Our goal is to measure the distance between two distributions $P_X$ and $P_Y$. However, instead of observing $X = \{x_i\}_i^n \sim P_X$, $Y = \{y_i\}_i^n \sim P_Y$, we observe $\epsilon$-contaminated $X'$ and $Y'$, where a possibly non-random $\epsilon$ proportion of one of the two variables is incorrectly grouped with the other for $0 < \epsilon < 1$. Without loss of generality,

we assume that the two samples have the same size $= n$ and that an $\epsilon$-proportion of $X$ is incorrectly grouped with $Y$. Specifically, let $C^* = \{c_i^*\}_i^m$, with $m = \lfloor \epsilon n \rfloor$ be the unobserved subset of $X$ that is grouped with $Y$. We can express the distributions over the observed samples in relation to the true distributions and the unknown contaminated samples as follows:

$$P_{Y'} = (1 - \alpha)P_Y + \alpha P_{C^*}$$
$$P_{X'} = (1 + \tilde{\alpha})P_X - \tilde{\alpha}P_{C^*},$$

where $\alpha = \epsilon/(1 + \epsilon)$ and $\tilde{\alpha} = \epsilon/(1 - \epsilon)$. We do not make any additional assumptions about $P_{C^*}$. Importantly, we do not assume that the contamination is random, meaning *we do not assume* that $P_{C^*} = P_{X'} = P_X$.

We assume that the value of $\epsilon$ is known *a priori*, or can be empirically estimated from other data sources. However, in section 5.5, we conduct a sensitivity analysis to examine the performance of our approach and others under violations of this assumption. We use $\mathbb{E}_{P_A}[A]$ to denote the expectation of $A$ according to the distribution $P_A(A)$, $A \cup B$ to denote the union of the set $A$ and $B$, and $A \setminus B$ to denote the difference between the two sets $A$ and $B$. We use $\#(A)$ to denote the cardinality of the set $A$. We use $\mathcal{X}'$ and $\mathcal{Y}'$ to denote the support of $X'$ and $Y'$ respectively.

We focus on the MMD as a measure of distance between distributions [Gretton et al., 2012]:

**Definition 1** *For $Z \sim P_Z$, $Z' \sim P_{Z'}$, $\mathcal{F}$ such that $\mathcal{F} : \mathcal{Z} \rightarrow \mathbb{R}$, and $k : \mathcal{Z} \times \mathcal{Z} \rightarrow \mathbb{R}$ with $k$ being a positive definite kernel matrix, the MMD is defined as*

$$\text{MMD}(\mathcal{F}, P_Z, P_{Z'}) = \sup_{f \in \mathcal{F}} \left( \mathbb{E}_{P_Z} f(Z) - \mathbb{E}_{P_{Z'}} f(Z') \right),$$

*and the witness function $f^*$ is defined as the function attaining the supremum in expression above, with $f^*(t) = \mathbb{E}_{P_Z}[k(Z, t)] - \mathbb{E}_{P_{Z'}}[k(Z', t)]$, up to a normalization constant.*

When $\mathcal{F}$ is set to be a general reproducing kernel Hilbert space (RKHS), the MMD defines a metric on probability distributions, and is equal to zero if and only if $P_Z = P_{Z'}$. Throughout, we fix $\mathcal{F}$ to be the RKHS with $\|f\|_{\mathcal{F}} \leq 1$ for all $f \in \mathcal{F}$ and drop $\mathcal{F}$ from the MMD arguments to simplify notation. We use $k(z, z')$ to denote the reproducing kernel of $\mathcal{F}$, and assume that $0 \leq k(x', y') \leq \kappa$ for all $x', y' \in \mathcal{X}', \mathcal{Y}'$.

Gretton et al. [2012], showed that when there is no measurement error, the following empirical estimate of the MMD is unbiased:

$$\widehat{\text{MMD}}(X, Y) = \frac{1}{n(n-1)} \sum_{i, j \neq i} k(x_i, x_j) \quad (1)$$

$$+ \frac{1}{n(n-1)} \sum_{i, j \neq i} k(y_i, y_j) - \frac{2}{n^2} \sum_{i, j} k(x_i, y_i). \quad (2)$$

As we show in the appendix section E, in the $\epsilon$-contamination setting, $\widehat{\mathrm{MMD}}$ is not guaranteed to be an unbiased estimate, meaning $\widehat{\mathrm{MMD}}(X', Y')$ might not converge to $\mathrm{MMD}(P_{X'}, P_{Y'})$. So instead we study partial identifiability of $\mathrm{MMD}(P_X, P_Y)$. Meaning, our goal is to estimate credible and informative lower and upper bounds on the unknown $\mathrm{MMD}(P_X, P_Y)$. For those bounds to be informative, they should be *sharp*, meaning they cannot be made tighter without any additional assumptions.

## 3 THEORY

Our goal is to estimate upper and lower bounds that reflect our uncertainty in the MMD due to measurement error.

To proceed with our analysis, it is helpful to parameterize the MMD as function of the contaminated samples $C$. With some abuse of notation, for an arbitrary distribution $P_C$, we have that:

$$\mathrm{MMD}(P_C, P_{X'}, P_{Y'}) = \sup_{f \in \mathcal{F}} \big[ (1-\epsilon) \mathbb{E}_{P_{X'}} f(X')$$
$$- (1+\epsilon) \mathbb{E}_{P_{Y'}} f(Y') + 2\epsilon \mathbb{E}_{P_C} f(C) \big], \quad (3)$$

with $\mathrm{MMD}(P_X, P_Y) = \mathrm{MMD}(P_{C^*}, P_{X'}, P_{Y'})$. Our first result characterizes the sharpest possible bounds that can be attained without additional assumptions.

**Proposition 1** *Let $(\mathcal{Y}', \Omega)$ be a measurable space with $Y' \in \mathcal{Y}'$ and let $\mathcal{P}$ be all the probability distributions on $(\mathcal{Y}', \Omega)$. Define $\mathcal{P}(\alpha)$ to be all the possible probability distributions over the unknown $C^*$, i.e., $\mathcal{P}(\alpha) = \{ (P_{Y'}(Y') - (1-\alpha)\varphi)/\alpha : \varphi \in \mathcal{P} \}$, then the following bounds are sharp:*

$$\inf_{P_C \in \mathcal{P}(\alpha)} \mathrm{MMD}(P_C, P_{X'}, P_{Y'}) \leq \mathrm{MMD}(P_{C^*}, P_{X'}, P_{Y'})$$
$$\leq \sup_{P_C \in \mathcal{P}(\alpha)} \mathrm{MMD}(P_C, P_{X'}, P_{Y'}),$$

The proofs for proposition 1 and all other statements are presented in the appendix. The intuition for proposition 1 is simple: without any additional assumptions, $C^*$ can take on any values in $\mathcal{Y}'$, and hence its corresponding distribution can be any distribution consistent with the observed data (i.e., any distribution $\in \mathcal{P}(\alpha)$). This means that the sharpest possible upper (lower) bound must be defined with respect to distributions over $P_C$ that maximize (minimize) the MMD.

We use $P_{\overline{C}}$ to denote the distribution that maximizes the third term in proposition 1 and define $P_{\underline{C}}$ similarly. Proposition 1 gives us a recipe for constructing empirical bounds on the true $\mathrm{MMD}(P_{C^*}, P_{X'}, P_{Y'})$. To get an estimate of the upper bound, we need to identify the values of $C$ that render $X' \cup C$ and $Y' \setminus C$ most dissimilar. For a lower bound, we need to identify values of $C$ that render $X' \cup C$ and $Y' \setminus C$ most similar. Unless otherwise noted, we will

focus on the analysis of the upper bound of the MMD since the arguments for the lower bound are nearly identical.

We further expand the empirical version of equation 3 to isolate the terms that depend on $C$, which gives us the empirical objective to optimize. First, we define a weighted version of the empirical witness function,

$$\psi(C, X', Y') := \frac{(1-\epsilon)}{n} \sum_i \sum_j k(x_i', c_j)$$
$$- \frac{(1+\epsilon)}{n} \sum_i \sum_j k(y_i', c_j)$$
$$+ \frac{\epsilon}{n} \sum_i \sum_{j \neq i} k(c_i, c_j).$$

As we show in Lemma A1, in order to estimate $\mathrm{MMD}(P_{\overline{C}}, P_{X'}, P_{Y'})$, we first need to identify $\widehat{C}$:

$$\widehat{C} = \underset{C \in Y', \#(C)=m}{\arg \max} \psi(C, X', Y'). \quad (4)$$

Note that optimizing $\psi$ under a cardinality constraint in this manner is an NP-hard optimization problem. Instead, we analyze approximation strategies in two regimes: when $\epsilon$ can take on any value in [0,1] and when $\epsilon$ is sufficiently close to 0. Our analysis relies on analyzing the stability of the estimation algorithms [Bousquet and Elisseeff, 2002].

**Approximation strategy for $\epsilon \in [0, 1]$.** For any value of $\epsilon$, we can directly maximize equation 4. Noting that: $\max_C \psi(C \in Y', X', Y') \leq \max_C \psi(C \in \mathcal{Y}', X', Y')$, we can utilize, for example, iterative optimization algorithms to estimate an approximate $\widehat{C}$. Specifically,

$$\widehat{C}_\circ = \underset{C \in \mathcal{Y}', \#(C)=m}{\arg \max} \psi(C, X', Y'). \quad (5)$$

The difference between equation 4 and 5 is that 5 can return any value for $\widehat{C}_\circ \in \mathcal{Y}'$, whereas 4 requires that $\widehat{C}_\circ \in Y'$.

While many iterative optimization algorithms can be used to optimize equation 5, we follow Jitkrittum et al. [2016] in using Quasi-Newton methods such as the L-BFGS-B algorithm [Byrd et al., 1995]. For this reason we refer to this iterative optimization approach as the Quasi-Newton optimization **QNO** approach. We stress that our analysis holds for any valid optimization approach.

In proposition 2, we study how fast the estimate based on $\widehat{C}_\circ$ converges to the true upper bound.

**Proposition 2** *For* $\mathrm{MMD}(P_{\overline{C}}, P_{X'}, P_{Y'})$ *as defined in proposition 1,* $\widehat{C}_\circ$ *as defined in equation 5, with* $0 \leq k(x', y') \leq \kappa$ *for all* $x', y' \in \mathcal{X}', \mathcal{Y}'$, *we have that:*

$$P_{X', Y'}\left\{ |\mathrm{MMD}(P_{\overline{C}}, P_{X'}, P_{Y'}) - \widehat{\mathrm{MMD}}(\widehat{C}_\circ, X', Y')| \right.$$
$$\left. > b_0 + \varepsilon \right\} \leq 2 \exp\left( \frac{-\varepsilon^2 n}{b_1} \right),$$

*for* $b_0 = 4\sqrt{\kappa}(n^{-1/2} + \epsilon m)$ *and* $b_1 = 2\kappa((1-\epsilon)(1 - \epsilon + \epsilon m)^2 + (1+\epsilon)(1 + \epsilon + \epsilon m)^2)$.

The proposition shows that the rate of convergence of the empirical MMD defined with respect to $\widehat{C}_\circ$ to the sharp upper bound depends on the sample size, the value of $\epsilon$ and the size of the contaminated set $m$. As $\epsilon$ decreases, the estimated $\widehat{\mathrm{MMD}}(\widehat{C}_\circ, X', Y')$ converges faster to its population counterpart $\mathrm{MMD}(P_{\overline{C}}, P_{X'}, P_{Y'})$. At $\epsilon = 0$, we recover the convergence rate of the uncontaminated $\widehat{\mathrm{MMD}}$ (Gretton et al. [2012], theorem 7). As expected, as the sample size increases, the estimate gets closer to its population counterpart. However, the $\epsilon m$ term in the denominator of the exponent means that the rate of convergence depends unfavorably on the size of the contaminated sample. The next section addresses this.

**Approximation strategy for a sufficiently small $\epsilon$.** This approach relies on the fact that for a fixed $n$, and as $\epsilon \to 0$ the third term in equation 4 vanishes.

Specifically for $\epsilon \approx 0$:

$$\psi(C, X', Y') \approx \frac{(1-\epsilon)}{n} \sum_i \sum_j k(x'_i, c_j) -$$
$$\frac{(1+\epsilon)}{n} \sum_i \sum_j k(y'_i, c_j) = \frac{1}{m} \sum_i \hat{f}'(c_i). \quad (6)$$

where $\hat{f}'$ is a weighted version of the empirical estimate of the witness function definted with respect to the observed contaminated samples.

This means that for $\epsilon$ close to 0, maximizing $\psi$ is equivalent to computing the value of the witness function for every sample in $Y'$, and then taking the subset with the highest values to be the estimate of $\widehat{C}$. Consider the following estimate of $\widehat{C}$:

$$\widehat{C}_{\hat{\gamma}} = \{y' : \hat{f}'(y') \geq \hat{\gamma}\} \text{ with } \hat{\gamma} = q(\hat{f}'(Y'), 1 - \alpha), \quad (7)$$

where $q(\hat{f}'(Y'), 1-\alpha)$ is defined as the $1-\alpha$ quantile of $\hat{f}'(Y')$. That is, $q(\hat{f}'(Y'), 1-\alpha) = \inf\{\hat{f}'(y') \in \hat{f}'(Y') : (1-\alpha) < \mathrm{CDF}(\hat{f}'(y'))\}$. Equation 7 describes taking the $y'$ samples with weighted witness function values in the top $1 - \alpha$ quantile as the candidates for contaminated samples. Next, we show that $\widehat{C}_{\hat{\gamma}}$ is a valid estimate of $\overline{C}$.

**Proposition 3** *Let* $C_\gamma$ *be the solution to equation 7 as* $n \to \infty$. *For a sufficiently small* $\epsilon$, *we have that* $P_{C_\gamma} = P_{\overline{C}}$, *where* $P_{\overline{C}}$ *is defined as the distribution that maximizes the third term in proposition 1.*

While the full proof is stated in the appendix, we find it helpful to highlight the key insight behind proposition 3. The key insight here is that the distribution over $C_\gamma$ *stochastically dominates* any other distribution over $Y'$ with respect to the transformation $f'(Y')$. Meaning, there exists no other distribution over a subset of $Y'$ with measure $\alpha$ that can give a larger $\mathbb{E}_C[f'(C)]$ than $\mathbb{E}_{C_\gamma}[f'(C_\gamma)]$. We note in passing that this construction extends the classical seminal work by Horowitz and Manski [1995] on estimation of population means using contaminated data to nonparametric estimation of distances between distributions. We refer to this approach as the stochastic dominance (**SD**) approach.

It remains to show that the estimate of the MMD defined with respect to $\widehat{C}_{\hat{\gamma}}$ as estimated using a *finite sample* converges to the true upper bound. We do that in the following proposition.

**Proposition 4** *For* $\mathrm{MMD}(P_{\overline{C}}, P_{X'}, P_{Y'})$ *as defined in proposition 1,* $\widehat{C}_{\hat{\gamma}}$ *as defined in equation 7 and* $\kappa$ *such that* $0 \leq k(x, y) \leq \kappa$ *for all* $x, y \in \mathcal{X}$. *Then as for a sufficiently small* $\epsilon$:

$$P_{X', Y'}\left\{ |\mathrm{MMD}(P_{\overline{C}}, P_{X'}, P_{Y'}) - \widehat{\mathrm{MMD}}(\widehat{C}_{\hat{\gamma}}, X', Y')| \right.$$
$$\left. > b_0 + \varepsilon \right\} \leq 2 \exp\left( \frac{-\varepsilon^2 n}{b_1} \right)$$

*for* $b_0 = 4(\kappa/n)^{1/2}(1 + \epsilon)$ *and* $b_1 = 2\kappa\big((1-\epsilon)^3 + (1 + \epsilon)(1 + 3\epsilon)^2\big)$.

Proposition 4 shows that unlike QNO, SD avoids the unfavorable dependence on $m$ leading to faster convergence. Similar to proposition 2, at $\epsilon = 0$, we recover the convergence rate of the uncontaminated $\widehat{\mathrm{MMD}}$.

The key advantage of SD over QNO is that it reduces the problem of estimating $\widehat{C}$ to estimating the quantile of the univariate distribution, $P_{f'(Y')}$, which is a single scalar. By contrast, the iterative optimization-based approach needs to identify an $m \times d$ matrix, with $d$ being the dimension of the data. While helpful, the SD approach is limited by the fact that it is a valid approximation only for $\epsilon$ sufficiently close to 0. Next, we present our main approach that extends the SD approach making it valid for any value of $\epsilon$.

## 4   APPROACH

In this section, we describe our main approach to estimating tight and credible upper and lower bounds on the MMD.

**Algorithm 1** Our approach (S-SD) for estimating upper bounds

**Input:** $X', Y', \epsilon, S$
$\widehat{C} := \{\}, \alpha^{(s)} = \epsilon/(\epsilon + S)$
  **for** $s = 1 \ldots S$ **do**
    $X^{(s)} = X' \cup \widehat{C}, Y^{(s)} = Y' \setminus \widehat{C}$
    Compute $\hat{f}^{(s)}(Y^{(s)})$ as per equation 8
    $\hat{\gamma}_{(1-\epsilon)} = q(\hat{f}^{(s)}(Y^{(s)}), 1 - \alpha^{(s)})$
    $\widehat{C}^s = \{y^{(s)} : \hat{f}^{(s)}(y^{(s)}) \geq \hat{\gamma}_{(1-\epsilon)}\}$
    $\widehat{C} := \widehat{C} \cup \widehat{C}^s$
**return** $\widehat{\text{MMD}}(\widehat{C}, X', Y')$

**Algorithm 2** Our approach (S-SD) for estimating lower bounds

**Input:** $X', Y', \epsilon, S$
$\undertilde{C} := \{\}, \alpha^{(s)} = \epsilon/(\epsilon + S)$
  **for** $s = 1 \ldots S$ **do**
    $X^{(s)} = X' \cup \undertilde{C}, Y^{(s)} = Y' \setminus \undertilde{C}$
    Compute $\hat{f}^{(s)}(Y^{(s)})$ as per equation 8
    $\hat{\gamma}_{\epsilon} = q(\hat{f}^{(s)}(Y^{(s)}), \alpha^{(s)})$
    $\undertilde{C}^s = \{y^{(s)} : \hat{f}^{(s)}(y^{(s)}) \leq \hat{\gamma}_{\epsilon}\}$
    $\undertilde{C} := \undertilde{C} \cup \undertilde{C}^s$
**return** $\widehat{\text{MMD}}(\undertilde{C}, X', Y')$

Unless otherwise noted, we describe the estimation procedure for constructing the upper bound since the lower bound is nearly identical. Our strategy hinges on identifying $\widehat{C}$, an $m$-sized subset of $Y'$ which, when removed from $Y'$ and added to $X'$, would render $Y'$ most dissimilar to $X'$, giving us a valid estimate of the the upper bound on the unknown $\widehat{\text{MMD}}(C^*, X', Y')$. Estimating $\widehat{C}$ allows us to estimate $\widehat{\text{MMD}}(\widehat{C}, X', Y')$ in a straightforward manner: we can simply substitute $\widehat{C}$ for $C$ in the empirical version of equation 3.

Our main approach builds upon the SD approach studied in section 3 by addressing its main limitation: that it gives a valid estimate of $\widehat{C}_{\hat{\gamma}}$ only for $\epsilon$ sufficiently close to 0. Our approach overcomes this limitation by dividing the task of estimating $\widehat{C}_{\hat{\gamma}}$ into multiple, easier tasks each with an effective $\epsilon^{(s)}$ that is smaller than the true $\epsilon$. Specifically, we divide the estimation process into $S$ steps, in each step we estimate $\widehat{C}_{\hat{\gamma}^{(s)}}^{(s)}$, for $\epsilon^{(s)} = \epsilon/S$. Dividing the estimation into $S$ steps, with each step having $\epsilon/S$-contamination means that each step of the estimation process will have an effective $\epsilon$ that is close enough to 0 making equation 7 a valid approximation, and overcoming the main limitation of SD. In the step $s$ of our algorithm, we calculate $\widehat{C}_{\hat{\gamma}^{(s)}}^{(s)} = \{y' \in \widehat{Y}^{(s)} : \hat{f}^{(s)}(\widehat{Y}^{(s)}) \geq \hat{\gamma}^{(s)}\}$, for $\hat{\gamma}^{(s)} = q(\hat{f}^{(s)}(\widehat{Y}^{(s)}), 1 - \alpha^{(s)})$ for $\alpha^{(s)} = \epsilon^{(s)}/(1 + \epsilon^{(s)})$, where

$$\hat{f}^{(s)}(\widehat{Y}^{(s)}) = \left(1 - \frac{\epsilon}{S}\right) \frac{1}{n} \sum_i \sum_j k(\hat{x}_i^{(s)}, \hat{y}_j^{(s)})$$
$$- \left(1 + \frac{\epsilon}{S}\right) \frac{1}{n} \sum_i \sum_j k(\hat{y}_i^{(s)}, \hat{y}_j^{(s)}), \quad (8)$$

with $\widehat{Y}^{(s)} = Y' \setminus \{\widehat{C}_{\hat{\gamma}^{(1)}}^{(1)}, \widehat{C}_{\hat{\gamma}^{(2)}}^{(2)}, \ldots \widehat{C}_{\hat{\gamma}^{(s-1)}}^{(s-1)}\}$, and $\widehat{X}^{(s-1)} = X' \cup \{\widehat{C}_{\hat{\gamma}^{(1)}}^{(1)}, \widehat{C}_{\hat{\gamma}^{(2)}}^{(2)}, \ldots \widehat{C}_{\hat{\gamma}^{(s-1)}}^{(s-1)}\}$.

We refer to our Stepwise Stochastic Dominance based approach as **S-SD**. We summarize our procedure for estimating the upper and lower bounds in algorithms 1 and 2 respec-

tively. We use $\undertilde{C}$ to denote the counterpart of $\widehat{C}$ defined with respect to the lower bound.

We note that $S$ is a user-specified parameter that takes on values between 0 and $m$. In section 5.5 we give practical guidance on how to set $S$. Code for our approach and the experiments in section 5 is available on github.com/mymakar/mmd_uncertainty.

## 5 EXPERIMENTS

In this section, we (1) analyze the credibility and tightness of our approach and baselines under varying data dimensions, varying sample sizes, and varying values of $\epsilon$. In addition, (2) we examine the computational efficiency of our approach as it compares to baselines. Finally, (3) we examine the sensitivity of our approach to misspecification of $\epsilon$ and under varying number of steps $S$.

To analyze the credibility and the tightness of the bounds estimated using our approach, we compute the False Coverage Rate (FCR) and Mean Interval Width (MIW). For $L$ draws of $X', Y'$ each of size $(1 - \epsilon)n$ and $(1 + \epsilon)n$ respectively, the FCR and the MIW are defined as follows:

$$\text{FCR} = 1 - \frac{1}{L} \sum_i \mathbb{1}\{\widehat{\text{MMD}}(\undertilde{C}, X_i', Y_i')$$
$$\leq \widehat{\text{MMD}}(C^*, X_i', Y_i') \leq \widehat{\text{MMD}}(\widehat{C}, X_i', Y_i')\},$$
$$\text{MIW} = \frac{1}{L} \sum_i |\widehat{\text{MMD}}(\undertilde{C}, X', Y') - \widehat{\text{MMD}}(\widehat{C}, X', Y')|$$

**Ablations**. We study the following ablations of our approach: **(1) SD**: For $S = 1$, S-SD becomes the same as SD. The performance of SD compared to S-SD highlights the importance of splitting the estimation procedure into $S$ steps. **(2)** Stepwise-QNO (**S-QNO**): Follows the same steps outlined in algorithm 1, however, instead of estimating $\widehat{C}_{\hat{\gamma}}^{(s)}$ and $\undertilde{C}_{\hat{\gamma}}^{(s)}$ as a subroutine, it estimates $\widehat{C}_{\circ}^{(s)}$ and $\undertilde{C}_{\circ}^{(s)}$ following equation 4 using the L-BFGS-B optimization algorithm.

|  | MIMIC ($N=100, d=2$) | | FOREST ($N=100, d=54$) | | BIO ($N=72, d=7128$) | |
| Approach | FCR | MIW | FCR | MIW | FCR | MIW |
| --- | --- | --- | --- | --- | --- | --- |
| S-SD (Ours) | **0.0 ± (0.0)** | **0.137 ± (0.008)** | **0.0 ± (0.0)** | **0.088 ± (0.003)** | **0.1 ± (0.03)** | **0.075 ± (0.001)** |
| S-QNO | 0.08 ± (0.067) | 0.119 ± (0.006) | 0.02 ± (0.02) | 0.084 ± (0.004) | 1.0 ± (0.0) | 0.059 ± (0.001) |
| QNO | 0.58 ± (0.069) | 0.13 ± (0.006) | 0.62 ± (0.069) | 0.033 ± (0.006) | 1.0 ± (0.0) | 0.037 ± (0.001) |
| SD | 0.64 ± (0.068) | 0.082 ± (0.01) | 0.9 ± (0.042) | 0.027 ± (0.005) | 0.13 ± (0.034) | 0.069 ± (0.001) |
| SM | 0.66 ± (0.067) | 0.08 ± (0.01) | 0.9 ± (0.042) | 0.026 ± (0.004) | 0.82 ± (0.038) | 0.037 ± (0.001) |
| Bootstrap | 0.94 ± (0.034) | 0.048 ± (0.002) | 0.4 ± (0.069) | 0.034 ± (0.001) | 0.25 ± (0.043) | 0.036 ± (0.001) |

Table 1: MIW and FCR for all datasets at $\epsilon = 0.2$. Numbers in bold correspond to lowest FCR with smallest MIW. Standard errors (in parentheses) computed by averaging over 100 trials. Results show that our approach performs better than all other approaches when the sample size is small and the dimension is large. In easier settings, our performs comparably to S-QNO.

In each step $s$, this approach gives an estimate for an $m/S$ subset of candidate contaminated samples. This ablation study highlights the importance of using the SD approach as a subroutine. **(3) QNO**: Similar to S-QNO with $S = 1$.

**Baselines.** In addition to our main approach and the ablations, we investigate the following baselines: **(1)** Submodular optimization (**SM**): based on the approach suggested in Kim et al. [2016]. It estimates $\widehat{C}$ by converting equation 4 into a submodular function by adding a submodular regularizer. Specifically, it greedily selects samples which maximise the function, $\max_m \hat{f}'(C) + \log \det k(C, C)$, where $\hat{f}'(C)$ is the witness function defined with respect to $X'$ and $Y'$, and $\log \det k(C, C)$ is the log-determinant regularizer. **(2) Bootstrap**: a simple bootstrapping approach, which constructs bounds by resampling both observed groups with replacement and computing the MMD multiple times. The upper and lower bounds are then defined as the $(1 - \alpha)$-th and $\alpha$ quantiles respectively over the distribution of resampled MMD values. The bootstrap estimates are centered around the typical MMD estimate (equation 1), and hence they show how it behaves under $\epsilon$-contamination [1].

For our approach, baselines and ablations, we fix the kernel to be the radial basis kernel (RBF) and use the median heuristic on the contaminated samples to determine bandwidth. Unless otherwise noted, we set the number of steps $S$ for S-SD and S-QNO to be $S = \min(m, 10)$; we take this minimum for when the total number of contaminated samples is less than the total number of steps. We examine the performance of different values of $S$ in section 5.5.

**Setup.** Since the true value of the contaminated samples $C^*$ is unobserved in real datasets, we resort to semi-simulated data where $X, Y$ represent real data, but the contaminated samples are simulated. We examine the performance of our approach, ablations and baselines in two settings. First, is the nonrandom contamination setting. In this setting, we pick the data points that maximize the difference between the two distributions to be the true contaminated samples. Specif-

ically, we simulate contamination by randomly sampling $C^*$, a set of size $m$ from the $\min(2m, n)$ samples in $X$ with the largest witness function values, where the witness function here is defined with respect to the uncontaminated $X, Y$. We then create the observed samples $X' = X \setminus C^*$ and $Y' = Y \cup C^*$. Second, is the random contamination setting, where $C^*$ is sampled at random from $X$. Since the nonrandom contamination setting is more challenging, we present the results from that setting in the main text. Results from the random contamination setting are presented in the appendix. We define $N = \#(X) + \#(Y)$, the total number of samples, and consider 3 tasks corresponding to 3 datasets:

1. **FOREST**: A publicly available dataset containing measurements of 54 cartographic variables such as elevation and slope [Blackard, 1998]. We consider the task of measuring the distance between the distribution over cartographic properties of two forest types: Lodgepole Pine and Spruce-Fir. We simulate $\epsilon$ contamination by flipping an $\epsilon$ proportion of Lodgepole Pine ($n = 283, 301$) labels to Spruce-Fir ($n = 211, 840$).

2. **MIMIC**: A publicly available chest radiographs and corresponding clinical data with over 377,000 chest X-ray images and radiology reports [Johnson et al., 2019a,b, Goldberger et al., 2000]. Here, we consider the task of measuring the distance between pneumonia predictions across two race groups – a common task in the fairness literature. In this setting, the sensitive attribute is measured with $\epsilon$-contamination. We use 60% of the data for training the model, 20% for validation, and the remaining 20% for MMD estimation. We use the training and validation data to fine tune a Densenet-121 [Huang et al., 2016] that was pretrained on Imagenet [Deng et al., 2009]. After training the model, we obtain the 2-dimensional logit predictions of the 20% of the data held out for MMD estimation, and simulate $\epsilon$-contamination by changing an $\epsilon$ proportion of Black ($n = 3897$) patients to White ($n = 11293$).

3. **BIO**: Unlike the 2-dimensional MIMIC data and 54-dimensional FOREST data, in the third task we examine a more extreme case of high dimensional data with few samples. We use publicly available leukemia gene expression

---

[1] In the appendix, we explicitly show how the typical estimate of the MMD behaves with varying $\epsilon$

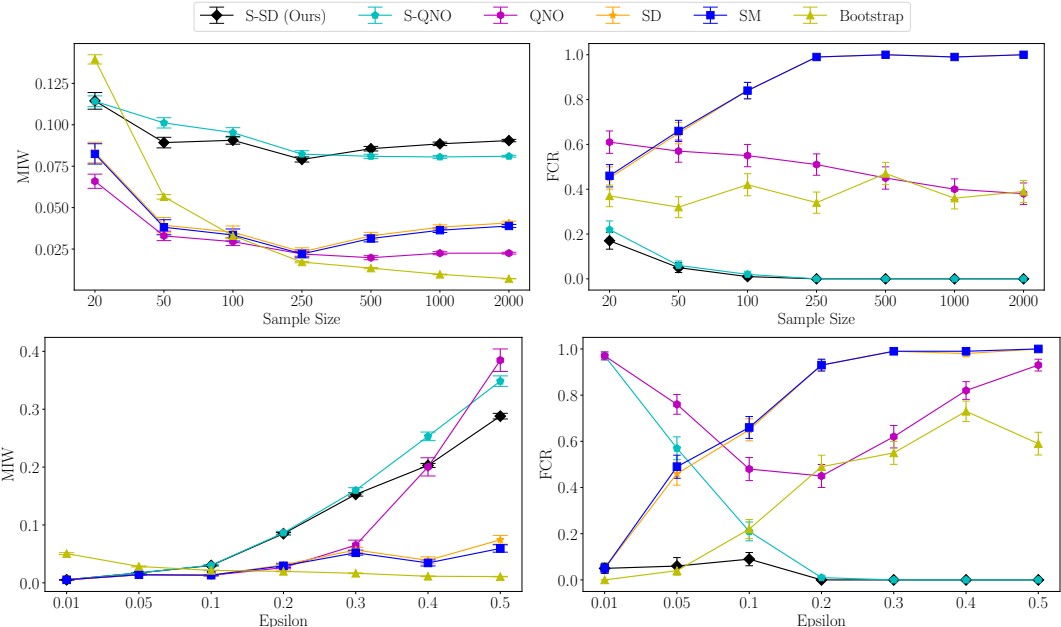

Figure 1: **Top**: Results on FOREST fixing $\epsilon = 0.2$ and increasing sample size from $N = 20$ to $N = 2000$. Bars indicate the SE of the FCR and MIW across all trials. As sample size increases, MIW decreases for all methods, with S-SD providing intervals with the lowest FCR for all sample sizes. **Bottom**: The MIW and FCR for each approach is shown as the intensity of $\epsilon$-contamination varies from $\epsilon = 0.01$ to $\epsilon = 0.5$ in FOREST ($N = 100, d = 54$). Bars indicate the SE of the FCR and MIW across all trials. As $\epsilon$ increases, S-SD reports tight and credible intervals for all values of $\epsilon$.

dataset (BIO) [Golub et al., 1999], which has 7128 measurements of gene expressions from DNA microarrays for 72 samples. The 72 samples are divided into binary groups of leukemia cancer cell types, acute lymphoblastic leukemia (ALL) and acute myeloid leukemia (AML), and we conduct the $\epsilon$ contamination by flipping $\epsilon$ of the ALL ($n = 47$) to AML ($n = 25$).

## 5.1 PERFORMANCE UNDER DIFFERENT DATA DIMENSIONS

In this section, we examine the effect of varying dimension. To do so, we compute the FCR and MIW of bounds estimated on MIMIC ($N = 100, d = 2$), FOREST ($N = 100, d = 54$), and BIO ($N = 72, d = 7128$) in table 1. We focus on the small sample regime as it is much more challenging. To get estimates for the standard error (SE) around the MIW and FCR, we repeat the experiment 100 times on 100 samples picked without replacement for MIMIC and FOREST. For BIO, we create 100 bootstrap samples. We fix $\epsilon = 0.2$, simulate contamination in 100 random samples, and calculate the upper and lower bounds for each approach.

The results in table 1 show that in all settings our approach gives the tightest (smallest MIW) and most credible (lowest FCR) estimates, while SD, QNO and S-QNO return bounds with a higher FCR. In settings where the dimensions are small, S-QNO performs significantly better than

QNO. However, both perform poorly when the dimension, $d$ is large. Such a finding makes sense: the stepwise algorithm reduces the dependence on the sample size, however the performance of both QNO and S-QNO appears to have some irreducible dependence on the dimension. This is not surprising, in BIO, for example, S-QNO is solving an optimization problem over an $m/S \times 7128$ parameter space, whereas S-SD is required to estimate the $(1-\alpha)/S$ quantile of a univariate distribution (that is the distribution over the values of the witness function). In this setting where $\epsilon = 0.2$, equation 7 is a poor approximation of equation 4, which explains the poor performance of SD. At $\epsilon = 0.2$ the typical estimate of the MMD (equation 1) is unreliable. Being centered around the typical estimate, Bootstrap is expected to give unreliable bounds. SM also performs poorly since it is designed to find *few* samples that explain the difference between the two corrupted distributions.

Overall, S-SD remains robust even in high dimensions, while other approaches do not. In the appendix, we repeat this experiment with $N = 2000$ for MIMIC and FOREST. The results are largely consistent with the findings presented here. However, as $N$ increases, the estimates for S-QNO in small dimensions become more comparable to S-SD.

For brevity, we present results on the FOREST dataset in the main text but include the similar analyses on MIMIC and BIO in the appendix.

## 5.2 PERFORMANCE UNDER DIFFERENT SAMPLE SIZES

We study the effect of increasing sample size. Fixing $\epsilon = 0.2$, we vary the sample size from $N = 20$ to $N = 2000$ by sampling from the FOREST dataset. For each sample size, we sample 100 times and compute the mean FCR and MIW and their corresponding standard errors. We plot the results for the MIW in figure 1 (top, left) and the FCR in figure 1 (top, right). The results show that the FCR for our approach, S-QNO and QNO decreases as the sample size increases revealing that these estimates are consistent. However, our approach gives the lowest FCR even in very small samples. In larger samples, S-QNO performs comparably to our approach. SD, SM and the bootstrap method all return overly conservative estimates that do not contain the true MMD.

## 5.3 PERFORMANCE UNDER DIFFERENT VALUES OF $\epsilon$

We investigate the effect of increasing contamination from $\epsilon = 0.01$ to $\epsilon = 0.9$. Similar to section 5.1, we focus on the small sample regime by fixing $N$ to be 100. We present the results here up to $\epsilon = 0.5$, and the rest in the appendix.

Figure 1 (bottom) shows that for small values of $\epsilon$, QNO and S-QNO perform poorly, giving high FCR. S-QNO resolves some of the issues by dividing the optimization into several steps, but still underperforms compared to our approach. SD gives a biased estimate of the bound for $\epsilon$ significantly higher than 0, as expected. Bootstrap gives valid bounds with low FCR only with near negligable values of $\epsilon$, where the typical MMD estimate is approximately valid.

The previous three experiments show that S-SD consistently gives credible and tight estimates of the upper and lower bounds on the value of the true MMD. Next, we examine the sensitivity of S-SD to the number of steps $S$.

## 5.4 COMPUTATIONAL EFFICIENCY

Next, we examine the computational efficiency of our approach as compared to baselines. Using the Forest dataset, with $n = 2000$, we vary the value of $\epsilon$ and measure the time in seconds that is required for each model to compute the upper and lower bounds. We repeat the experiment 100 times and report mean time and standard errors.

The results, shown in figure 2, indicate that our basic (non-stepwise) approach is the fastest, and particularly it is faster than the non-stepwise QNO approach while our main approach S-SD is faster than its counterpart, S-QNO. Importantly, the plot implies that increasing the value of $\epsilon$ has a negligible effect on the computational time of SD and our main approach S-SD.

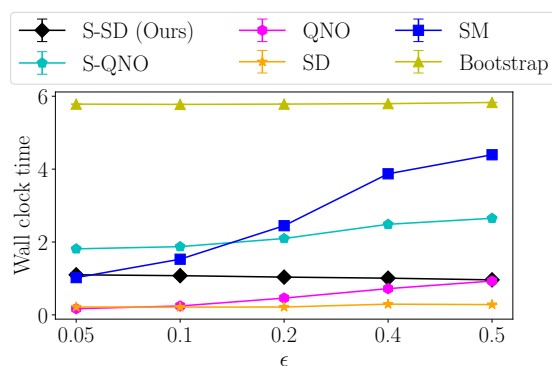

Figure 2: Computational efficiency: $x$-axis shows the value of $\epsilon$, $y$-axis shows the wall clock time in seconds. Our basic (non-stepwise) approach

## 5.5 SENSITIVITY ANALYSES

**Sensitivity to incorrect values of $\epsilon$.** Next, we examine the sensitivity of S-SD and other methods to incorrect values of $\epsilon$. To do so, we fix the true value of $\epsilon$ to be 0.1 but sweep $\tilde{\epsilon}$, the value given to each of the models, from 0.01 to 0.5. This means that the assumption of known/correct level of noise is only satisfied when $\tilde{\epsilon} = \epsilon = 0.1$. Similar to section 5.1, we focus on the small sample regime by fixing $N$ to be 100.

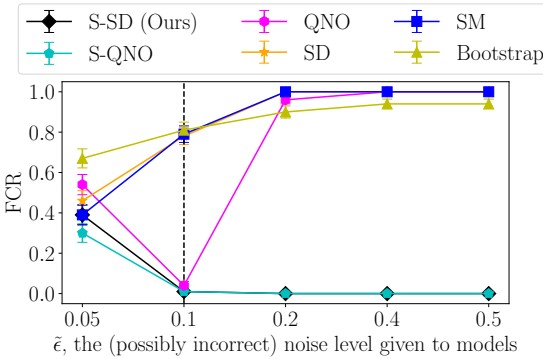

Figure 3: Sensitivity to incorrect values of $\epsilon$. True value of $\epsilon$ is 0.1. Models are given $\tilde{\epsilon}$ values shown on the $x$-axis

Figure 3 shows the value of $\tilde{\epsilon}$ on the $x$-axis and the corresponding FCR on the $y$-axis. In the appendix, we present a plot with the corrsponding mean interval widths for each level of $\tilde{\epsilon}$. The results show that only our two approaches (S-SD and S-QNO) achieve an FCR of 0 whenever $\tilde{\epsilon} \geq \epsilon$. This suggests a practical guideline: when in doubt, users should err on the side of a higher $\epsilon$ estimate with the trade-off of wider intervals (as reported in the appendix). Other methods do not give such a guarantee: they consistently give overly conservative intervals with poor coverage.

**Sensity to the number of steps.** We examine the sensitivity of S-SD to the number of steps $S$. To do so, we sample

| No. of Steps | S-SD (Ours) | |
| | FCR | MIW |
| --- | --- | --- |
| 2 | $0.21 \pm (0.091)$ | $0.082 \pm (0.001)$ |
| 3 | $0.13 \pm (0.034)$ | $0.079 \pm (0.001)$ |
| 5 | $0.0 \pm (0.0)$ | $0.088 \pm (0.001)$ |
| 10 | $0.0 \pm (0.0)$ | $0.08 \pm (0.001)$ |
| 20 | $0.0 \pm (0.0)$ | $0.091 \pm (0.001)$ |
| 50 | $0.0 \pm (0.0)$ | $0.091 \pm (0.001)$ |

Table 2: Varying number of steps for S-SD in FOREST ($N = 2000, d = 54$) with $\epsilon = 0.2$. Standard errors (in parentheses) over 100 trials. Results imply that setting $S$ to be large gives lower FCR.

$n = 2000$ from FOREST, vary the value of $S$, and examine the performance of our main approach, S-SD. We repeat the experiment 100 times using 100 different samples from FOREST, each of size 2000 to compute the standard errors around the FCR and MIW.

Table 2 shows the results. The results imply that we can get bound estimates that give a FCR of zero even with a very few number of steps. The MIW increase slightly and starts to plateau as the number of steps increases. This implies that a reasonable choice of $S$ to ensure a low FCR would be the largest possible value which does not lead to a computationally prohibitive number of iterations. Recall that there is a natural upper bound on $S = m$. In the appendix, we repeat this experiment for S-QNO showing similar robustness.

## 6 CONCLUSION

We studied the problem of comparing two distributions when the data is collected with some measurement error. Specifically, we showed that typical estimates of kernel based distances are unreliable when the data is measured with some $\epsilon$ contamination, where an $\epsilon$ proportion of one sample is erroneously included with the other. We showed both empirically and theoretically that a straightforward optimization approach to measuring uncertainty has an unfavorable dependence on the size of the contaminated set. Instead, we proposed a stepwise approach to estimate credible and tight upper and lower bounds and showed that it converges faster than alternatives to the true upper and lower bounds. Empirically, we showed that our approach outperforms all baselines. Looking beyond this work, it would be interesting to study other commonly occurring measurement error mechanisms and study their effect on measuring the MMD and other related estimates such as the Hilbert Schmidt independence criterion.

**Extensions of this work.** While beyond the scope of this work, it might be interesting to understand how our suggested approaches can be used in the context of hypothesis testing, where the goal is to formally test if the two distributions are similar. We note that such a test can be done by combining approaches for hypothesis testing using "interval test statistics" (see Kreinovich et al. [2008] for a summary) with approaches for acquiring empirical estimates of the MMD under the null distribution Gretton et al. [2009].

We also note that extending our approach to settings where both variables are contaminated is likely a trivial extension of our work. Specifically, it might be appropriate to conduct an iterative procedure where we find $\widehat{C}_x$: the samples observed in $Y'$ that are truly sampled from $P_X$ and then find $\widehat{C}_y$ the samples observed in $X'$ that are truly sampled from $P_Y$ iteratively until meeting some convergence criteria.

## Acknowledgements

We are thankful for feedback from anonymous reviewers. The work was done when Ron Nafshi was a student at the University of Michigan. This material is based upon work supported by the National Science Foundation under Grant No.2153083 and 2337529. Any opinions, findings, and conclusions or recommendations expressed in this material are those of the author(s) and do not necessarily reflect the views of the National Science Foundation.

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

# Partial identification of the maximum mean discrepancy with mismeasured data (Appendix)

**Ron Nafshi**[1]  **Maggie Makar**[2]

[1]Fintica AI, Tel Aviv, Israel
[2]University of Michigan, Ann Arbor, MI

## A  PROOF FOR PROPOSITION 1

**Proposition A1 (Restated Proposition 1 in the main text)** *Let $(\mathcal{Y}', \Omega)$ be a measurable space with $Y' \in \mathcal{Y}'$ and let $\mathcal{P}$ be all the probability distributions on $(\mathcal{Y}', \Omega)$. Define $\mathcal{P}(\alpha)$ to be all the possible probability distributions over the unknown $C^*$, i.e., $\mathcal{P}(\alpha) = \{(P_{Y'}(Y') - (1-\alpha)\varphi)/\alpha : \varphi \in \mathcal{P}\}$, then the following bounds are sharp:*

$$\inf_{P_C \in \mathcal{P}(\alpha)} \text{MMD}(P_C, P_{X'}, P_{Y'}) \leq \text{MMD}(P_{C^*}, P_{X'}, P_{Y'}) \leq \sup_{P_C \in \mathcal{P}(\alpha)} \text{MMD}(P_C, P_{X'}, P_{Y'}),$$

*Proof.* Consider the upper bound, $\sup_{P_C \in \mathcal{P}(\alpha)} \text{MMD}(P_C, P_{X'}, P_{Y'})$, and let $P_{\overline{C}} = \arg\sup_{P_C \in \mathcal{P}(\alpha)} \text{MMD}(P_C, P_{X'}, P_{Y'})$. Note that without additional assumptions, it is possible that $P_{C^*} = P_{\overline{C}}$. In this case, the upper bound holds with equality. I.e., $\text{MMD}(P_{C^*}, P_{X'}, P_{Y'}) = \sup_{P_C \in \mathcal{P}(\alpha)} \text{MMD}(P_C, P_{X'}, P_{Y'})$. Hence the upper bound is sharp. A similar argument can be constructed to show that the lower bound is sharp.

## B  PROOF FOR PROPOSITION 2

Before proceeding to the main proof, we restate the following definition from Gretton et al. [2012].

**Definition A1 (Restated definition 30 in Gretton et al. [2012])** *. Let $\mathcal{F}$ be the unit ball in an RKHS, with kernel bounded according to $0 \leq k(x, y) \leq \kappa$. Let $Z$ be an i.i.d. sample of size $n$ drawn according to a probability measure $P_Z$ and let $\sigma_i$ be i.i.d and take values in $\{-1, 1\}$ with equal probability and $\boldsymbol{\sigma} = \{\sigma_i\}_{i=1}^n$. We define the Rademacher average:*

$$\mathcal{R}_n(\mathcal{F}, Z) = \mathbb{E}_{\boldsymbol{\sigma}} \sup_{f \in \mathcal{F}} \left| \frac{1}{n} \sum_i f(z_i) \right| \leq \left( \frac{\kappa}{n} \right)^{1/2}$$

**Proposition A2 (Restated Proposition 2 in the main text)** *For $\text{MMD}(P_{\overline{C}}, P_{X'}, P_{Y'})$ as defined in proposition 1, $\widehat{C}_\circ$ as defined in equation 5, with $\#(\widehat{C}_\circ) = m$, $0 \leq k(x', y') \leq \kappa$ for all $x', y' \in \mathcal{X}', \mathcal{Y}'$, we have that:*

$$P_{X',Y'}\left\{ |\text{MMD}(P_{\overline{C}}, P_{X'}, P_{Y'}) - \widehat{\text{MMD}}(\widehat{C}_\circ, X', Y')| > b_0 + \varepsilon \right\} \leq 2\exp\left( \frac{-\varepsilon^2 n}{b_1} \right),$$

*for $b_0 = 4\sqrt{\kappa}(n^{-1/2} + \epsilon m)$ and $b_1 = 2\kappa((1-\epsilon)(1-\epsilon+\epsilon m)^2 + (1+\epsilon)(1+\epsilon+\epsilon m)^2)$*

*Proof.* Define $\hat{c}_i^\circ$ such that $\widehat{C}_\circ = \{\hat{c}_i^\circ\}_{i=1}^m$ and consider the absolute difference term:

$$|\text{MMD}(P_{\overline{C}}, P_{X'}, P_{Y'}) - \widehat{\text{MMD}}(\widehat{C}_\circ, X', Y')|$$
$$= \left| \sup_{f \in \mathcal{F}} \left[ (1-\epsilon)\mathbb{E}_{P_{X'}} f(X') - (1+\epsilon)\mathbb{E}_{P_{Y'}} f(Y') + 2\epsilon\mathbb{E}_{\overline{C}} f(\overline{C}) \right] \right.$$

$$- \sup_{f \in \mathcal{F}} \Big[ \frac{(1-\epsilon)}{n} \sum_i f(x_i') - \frac{(1+\epsilon)}{n} \sum_i f(y_i') + \frac{2\epsilon}{n} \sum_i f(\hat{c}_i^\circ) \Big] \Big|$$

$$\leq \sup_{f \in \mathcal{F}} \Big| (1-\epsilon) \mathbb{E}_{P_{X'}} f(X') - (1+\epsilon) \mathbb{E}_{P_{Y'}} f(Y') + 2\epsilon \mathbb{E}_{\overline{C}} f(\overline{C})$$

$$- \frac{(1-\epsilon)}{n} \sum_i f(x_i') + \frac{(1+\epsilon)}{n} \sum_i f(y_i') - \frac{2\epsilon}{n} \sum_i f(\hat{c}_i^\circ)) \Big|$$

$$:= \Delta(X', Y', P_{X'}, P_{Y'})$$

We will next attempt to bound the difference between $\Delta_{\mathcal{D}}(P_{X'}, P_{Y'}, X', Y')$ and its expectation by applying McDiarmid's inequality. To do so, we first need to verify that $\Delta_{\mathcal{D}}(P_{X'}, P_{Y'}, X', Y')$ satisfies the bounded difference property. We do so in two steps. In the first step, we consider the case where we replace one of the $X'$ samples. Specifically, we consider the data $\mathcal{D}_{\pi j}^{X'} = \{X'_{\pi j}, Y'\}$, where $X'_{\pi j} = \{x_1', x_2', \dots, x_{i-1}', x_j', x_{i+1}', \dots x_{(1-\epsilon)n}'\}$. Let $\widetilde{C}_\circ$ denote the estimate of $\widehat{C}$ according to equation 5 using $\mathcal{D}_{\pi j}^{X'}$ rather than $\mathcal{D}$. In that case, we have that:

$$|\Delta_{\mathcal{D}}(P_{X'}, P_{Y'}, X', Y') - \Delta_{\mathcal{D}_j^{X'}}(P_{X'}, P_{Y'}, X'_{\pi j}, Y')|$$

$$\leq \sup_f \Big| \frac{(1-\epsilon)}{n} (\sum_i f(x_i') - f(x_i') + f(x_j')) - \frac{(1+\epsilon)}{n} \sum_i f(y_i')$$

$$+ \frac{2\epsilon}{n} \sum_i f(\tilde{c}_i^\circ) - \frac{(1-\epsilon)}{n} \sum_i f(x_i') + \frac{(1+\epsilon)}{n} \sum_i f(y_i') - \frac{2\epsilon}{n} \sum_i f(\hat{c}_i^\circ) \Big|$$

$$\leq \sup_f \Big| \frac{(1-\epsilon)}{n} (-f(x_i') + f(x_j')) + \frac{2\epsilon}{n} \sum_i f(\tilde{c}_i^\circ) - \frac{2\epsilon}{n} \sum_i f(\hat{c}_i^\circ) \Big|$$

$$\leq \frac{(1-\epsilon)}{n} (\sup_f |f(x_i')| + \sup_f |f(x_j')|) + \frac{2\epsilon}{n} \sup_f (\sum_i f(\tilde{c}_i^\circ) - \sum_i f(\hat{c}_i^\circ))$$

$$\leq \frac{(1-\epsilon)}{n} (2\sqrt{\kappa}) + \frac{2\epsilon}{n} (m\sqrt{\kappa}) = \frac{2\sqrt{k}}{n} (1 - \epsilon + \epsilon m) \tag{9}$$

Second, we consider the case where we replace one of the $Y'$ samples. Specifically, we consider the data $\mathcal{D}_{\pi j}^{Y'} = \{X', Y'_{\pi j}\}$, where $Y'_{\pi j} = \{y_1', y_2', \dots, y_{i-1}', y_j', y_{i+1}', \dots y_{(1+\epsilon)n}'\}$. In that case, by a similar construction to the previous case, we have that:

$$|\Delta_{\mathcal{D}}(P_{X'}, P_{Y'}, X', Y') - \Delta_{\mathcal{D}_j^{Y'}}(P_{X'}, P_{Y'}, X', Y'_{\pi j})| \leq \frac{2\sqrt{k}}{n} (1 + \epsilon + \epsilon m) \tag{10}$$

Combining the results from equations 9 and 10, we can apply McDiarmid with denominator:

$$(1-\epsilon)n \Big( \frac{2\sqrt{k}}{n} (1 - \epsilon + \epsilon m) \Big)^2 + (1+\epsilon)n \Big( \frac{2\sqrt{k}}{n} (1 + \epsilon + \epsilon m) \Big)^2$$

$$= \frac{4\kappa}{n} \Big( (1-\epsilon)(1 - \epsilon + \epsilon m)^2 + (1+\epsilon)(1 + \epsilon + \epsilon m)^2 \Big).$$

I.e.,:

$$P_{X', Y'} \Big\{ \Delta_{\mathcal{D}}(P_{X'}, P_{Y'}, X', Y') - \mathbb{E}_{X', Y'} \Big[ \Delta_{\mathcal{D}}(P_{X'}, P_{Y'}, X', Y') \Big] > \varepsilon \Big\} \leq 2 \exp \Big( \frac{-\varepsilon^2 n}{b_1} \Big), \tag{11}$$

where $b_1 = 2\kappa((1-\epsilon)(1 - \epsilon + \epsilon m)^2 + (1+\epsilon)(1 + \epsilon + \epsilon m)^2)$.

It remains to control $\mathbb{E}_{X', Y'} \Big[ \Delta_{\mathcal{D}}(P_{X'}, P_{Y'}, X', Y') \Big]$. To do so we use the $\beta$-stability property and symmetrization Van Der Vaart et al. [1996]. We note that the $\beta$-stability of the hypothesis is a direct consequence of the boundedness of $k(.,.)$ by $\kappa$. Let $X^\bullet$ and $Y^\bullet$ be i.i.d samples of sizes $(1-\epsilon)n$ and $(1+\epsilon)n$ respectively, we have that:

$$\mathbb{E}_{X', Y'} \Big[ \Delta_{\mathcal{D}}(P_{X'}, P_{Y'}, X', Y') \Big]$$

$$= \mathbb{E}_{X',Y'} \sup_f \left| (1-\epsilon)\mathbb{E}_{P_{X'}} f(X') - (1+\epsilon)\mathbb{E}_{P_{Y'}} f(Y') + 2\epsilon\mathbb{E}_{\overline{C}} f(\overline{C}) \right.$$

$$\left. - \frac{(1-\epsilon)}{n} \sum_i f(x_i') + \frac{(1+\epsilon)}{n} \sum_i f(y_i') - \frac{2\epsilon}{n} \sum_i f(\hat{c}_i^\circ)) \right|$$

$$= \mathbb{E}_{X',Y'} \sup_f \left| (1-\epsilon)\mathbb{E}_{X^\bullet}\left(\frac{1}{n}\sum_i f(x_i^\bullet)\right) - \frac{1-\epsilon}{n}\sum_i f(x_i') - (1+\epsilon)\mathbb{E}_{Y^\bullet}\left(\frac{1}{n}f(y_i^\bullet)\right) + \frac{1+\epsilon}{n}\sum_i f(y_i') \right.$$

$$\left. + 2\epsilon\mathbb{E}_{X^\bullet,Y^\bullet}\left(\frac{1}{n}f(\dot{c}_i^\circ)\right) - \frac{2\epsilon}{n}\sum_i f(\hat{c}_i^\circ) \right|$$

$$\leq \mathbb{E}_{X',Y',X^\bullet,Y^\bullet} \sup_f \left| \frac{1-\epsilon}{n}\sum_i f(x_i^\bullet) - \frac{1-\epsilon}{n}\sum_i f(x_i') - \frac{1+\epsilon}{n}\sum_i f(y_i^\bullet) + \frac{1+\epsilon}{n}\sum_i f(y_i') \right.$$

$$\left. + \frac{2\epsilon}{n}\sum_i f(\dot{c}_i^\circ) - \frac{2\epsilon}{n}\sum_i f(\hat{c}_i^\circ) \right|$$

$$\leq \mathbb{E}_{X',Y',X^\bullet,Y^\bullet} \sup_f \left| \frac{1-\epsilon}{n}\sum_i f(x_i^\bullet) - \frac{1-\epsilon}{n}\sum_i f(x_i') - \frac{1+\epsilon}{n}\sum_i f(y_i^\bullet) + \frac{1+\epsilon}{n}\sum_i f(y_i') \right|$$

$$+ \mathbb{E}_{X',Y',X^\bullet,Y^\bullet} \sup_f \left| \frac{2\epsilon}{n}\sum_i f(\dot{c}_i^\circ) - \frac{2\epsilon}{n}\sum_i f(\hat{c}_i^\circ) \right|$$

$$\leq \mathbb{E}_{X',Y',X^\bullet,Y^\bullet,\sigma',\sigma^\bullet} \sup_f \left| \frac{1-\epsilon}{n}\sum_i \sigma_i'(f(x_i^\bullet) - f(x_i')) + \frac{1+\epsilon}{n}\sum_i \sigma_i^\bullet(f(y_i^\bullet) - f(y_i')) \right|$$

$$+ \sup_{X',Y',X^\bullet,Y^\bullet} \left| \frac{2\epsilon}{n}\sum_i f(\dot{c}_i^\circ) - \frac{2\epsilon}{n}\sum_i f(\hat{c}_i^\circ) \right|$$

$$\leq \mathbb{E}_{X',X^\bullet,\sigma} \sup_f \left| \frac{1-\epsilon}{n}\sum_i \sigma_i'(f(x_i^\bullet) - f(x_i')) \right| + \mathbb{E}_{Y',Y^\bullet,\sigma} \sup_f \left| \frac{1+\epsilon}{n}\sum_i \sigma_i^\bullet(f(y_i^\bullet) - f(y_i')) \right|$$

$$+ \frac{2\epsilon}{n} \sup_{X',Y',X^\bullet,Y^\bullet} \left| \sum_i f(\dot{c}_i^\circ) - \sum_i f(\hat{c}_i^\circ) \right|$$

$$\leq 2[(1-\epsilon)\mathcal{R}_n(\mathcal{F},X') + (1+\epsilon)\mathcal{R}_n(\mathcal{F},Y')] + 2\epsilon m\sqrt{\kappa}]$$

$$\leq 2[(1-\epsilon)(\kappa/n)^{1/2} + (1+\epsilon)(\kappa/n)^{1/2} + 2\epsilon m\kappa^{1/2}]$$

$$\leq 4\sqrt{\kappa}(n^{-1/2} + \epsilon m).$$

Substituting $4\sqrt{\kappa}(n^{-1/2} + \epsilon m)$ for $\mathbb{E}_{X',Y'}\left[\Delta_\mathcal{D}(P_{X'}, P_{Y'}, X', Y')\right]$ in equation 11 gives the desired result.

## C  PROOF FOR PROPOSITION 3

Before stating the main proof, we begin by outlining the following definition, and lemmas.

**Definition A2** *Random variable $Z$ has first-order stochastic dominance (or stochastic dominance for short) over random variable $Z'$ if for any outcome $t$, $Z$ gives at least as high a probability of receiving at least $t$ as does $Z'$, and for some $t$, $Z$ gives a higher probability of receiving at least $t$.*

**Lemma A1** *Let $(\mathcal{Y}', \Omega)$ be a measurable space with $Y' \in \mathcal{Y}'$, and let $\mathcal{P}$ be all the probability distributions on $(\mathcal{Y}', \Omega)$. For $\mathcal{P}(\alpha) = \{(P_{Y'}(Y') - (1-\alpha)\varphi)/\alpha : \varphi \in \mathcal{P}\}$. We have that*

$$\arg\sup_{P_C \in \mathcal{P}(\alpha)} \mathrm{MMD}(P_C, P_{X'}, P_{Y'}) = \arg\sup_{P_C \in \mathcal{P}(\alpha)} \mathbb{E}_{P_C}[\tilde{f}'(C)],$$

*where*

$$\tilde{f}'(C) = (1-\epsilon)\mathbb{E}_{P_{X'}}[k(C, X')] - (1+\epsilon)\mathbb{E}_{P_{Y'}}[k(C, Y')] + \epsilon\mathbb{E}_{P_C} k(C, C) \tag{12}$$

*Proof.* The proof is a straight forward derivation from the definition of the MMD and the witness function. We present the derivation below, with all $\sup_{P_C}$ to be understood as $\sup_{P_C \in \mathcal{P}(\alpha)}$. We use $\widetilde{X}$ to denote $X' \cup C$ and $\widetilde{Y}$ to denote $Y' \setminus C$ for an arbitrary $C$.

$$\arg\sup_{P_C}\left[\text{MMD}(P_C, P_{X'}, P_{Y'})\right]$$

$$= \arg\sup_{P_C}\left[\sup_{f\in\mathcal{F}}\left[\mathbb{E}_{P_{\widetilde{X}}}[f(\widetilde{X})] - \mathbb{E}_{P_{\widetilde{Y}}}[f(\widetilde{Y})]\right]\right]$$

$$= \arg\sup_{P_C}\left[\mathbb{E}_{P_{\widetilde{X}}}[k(\widetilde{X}, \widetilde{X})] - \mathbb{E}_{P_{\widetilde{X}}}\mathbb{E}_{P_{\widetilde{Y}}}[k(\widetilde{X}, \widetilde{Y})] - \mathbb{E}_{P_{\widetilde{X}}}\mathbb{E}_{P_{\widetilde{Y}}}[k(\widetilde{X}, \widetilde{Y})] + \mathbb{E}_{P_{\widetilde{Y}}}[k(\widetilde{Y}, \widetilde{Y})]\right]$$

$$= \arg\sup_{P_C}\Big[(1-\epsilon)^2\mathbb{E}_{P_{X'}}[k(X', X')] + (1+\epsilon)^2\mathbb{E}_{P_{y'}}[k(y', y')]$$
$$- 2(1+\epsilon)(1-\epsilon)\mathbb{E}_{P_{X'}}\mathbb{E}_{P_{Y'}}[k(X', Y')] + 4\epsilon\big((1-\epsilon)\mathbb{E}_{P_C}\mathbb{E}_{P_{X'}}[k(C, X')]$$
$$- (1+\epsilon)\mathbb{E}_{P_C}\mathbb{E}_{P_{Y'}}[k(C, Y')] + \mathbb{E}_{P_C}\mathbb{E}_{P_C}[k(C, C)]\big)\Big]$$

$$= \arg\sup_{P_C}\left[\mathbb{E}_{P_C}\left[(1-\epsilon)\mathbb{E}_{P_{X'}}[k(C, X')] - (1+\epsilon)\mathbb{E}_{P_{Y'}}[k(C, Y')] + \mathbb{E}_{P_C}[k(C, C)]\right]\right]$$

$$= \arg\sup_{P_C}\left[\tilde{f}'(C)\right],$$

which completes the proof.

Note that the empirical version of equation 12 corresponds to equation 4 in the main text.

**Corollary A1** *Under the same conditions as Lemma A1, and for a sufficiently small $\epsilon$, we have that*

$$\arg\sup_{P_C\in\mathcal{P}(\alpha)}\text{MMD}(P_C, P_{X'}, P_{Y'}) \gtrless \arg\sup_{P_C\in\mathcal{P}(\alpha)}\mathbb{E}_{P_C}[f'(C)],$$

*where*

$$f'(C) = (1-\epsilon)\mathbb{E}_{P_{X'}}[k(C, X')] - (1+\epsilon)\mathbb{E}_{P_{Y'}}[k(C, Y')]$$

*Proof.* The proof directly follows from Lemma A1 and the fact that for a sufficiently small $\epsilon$, we have that $f'(C) \approx \tilde{f}'(C)$.

**Proposition A3 (Restated proposition 3 from the main text)** *Let $C_\gamma$ be the solution to equation 7 as $n \to \infty$. For a sufficiently small $\epsilon$, we have that $P_{C_\gamma} = P_{\overline{C}}$, where $P_{\overline{C}}$ is defined as the distribution that maximizes the third term in proposition 1.*

*Proof.* Recall that:

$$P_{Y'}(Y') = (1-\alpha)P_Y(Y) + \alpha P_{C^*}(C^*),$$

and note that the kernel $k$ is a measurable mapping, hence $f'$ is also a measurable mapping. This implies that $f'(Y')$ is measurable with respect to $Y'$ and we can express the distribution over $f'(Y')$. Letting $Q_{Y'} := P_{Y'}(f'(Y'))$, $Q_Y := P_Y(f'(Y))$, and $Q_{C^*} := P_{C^*}(f(C^*))$, we have that:

$$Q_{Y'}(Y') = (1-\alpha)Q_Y(Y) + \alpha Q_{C^*}(C^*).$$

Using the notation $Q_{Y'}[-\infty, t]$ to denote the cumulative distribution function (CDF) of $Q_{Y'}(Y')$ from values $-\infty$ to $t$, we can write the CDF over $C_\gamma$ as the CDF of a truncated distribution, which gives us the following:

$$Q_{C_\gamma}[-\infty, t] = \begin{cases} 0 & \text{if } t < \gamma \\ \left(Q_{Y'}[-\infty, t] - (1-\alpha)\right)/\alpha & \text{if } t \geq \gamma. \end{cases}$$

Consider the following distribution:

$$\varphi_0[-\infty, t] = \begin{cases} Q_{Y'}[-\infty, t]/(1-\alpha) & \text{if } t < \gamma \\ 1 & \text{if } t \geq \gamma. \end{cases}$$

Note that:

$$(1-\alpha)\varphi_0[-\infty,t] + \alpha Q_{C_\gamma}[-\infty,t] = Q_{Y'}[-\infty,1]$$

which means that $Q_{C_\gamma} \in \mathcal{P}(\alpha)$. Next we will make the argument that $Q_{C_\gamma}$ stochastically dominates all other distributions in $\mathcal{P}(\alpha)$. Note that for any $\varphi_1$, if $t < \gamma$

$$Q_{C_\gamma}[-\infty,t] - \varphi_1[-\infty,t] = 0 - \varphi_1[-\infty,t] \le 0.$$

However, suppose that there exists some $\varphi_1 \in \mathcal{P}(\alpha)$, and that it stochastically dominates $Q_{C_\gamma}$. I.e., for $t \ge \gamma$:

$$\varphi_1[-\infty,t] < Q_{C_\gamma}[-\infty,t]$$
$$\Rightarrow \varphi_1[-\infty,t] < \Big(Q_{Y'}[-\infty,t] - (1-\alpha)\Big)/\alpha$$
$$\Rightarrow \alpha\varphi_1[-\infty,t] < Q_{Y'}[-\infty,t] - (1-\alpha),$$

Hence we have that $(1-\alpha)\varphi + \alpha\varphi_1 < Q_{Y'}[-\infty,1]$ for all $\varphi \in \mathcal{P}$, which implies that $\varphi_1 \notin \mathcal{P}(\alpha)$, which is a contradiction.

This shows that $Q_{C_\gamma}[-\infty,t]$ stochastically dominates all distributions in $\mathcal{P}(\alpha)$, which means that:

$$\mathbb{E}_{Q_{C_\gamma}}[f'(C_\gamma)] > \mathbb{E}_{Q_C}[f'(C)]$$
$$\Rightarrow \mathbb{E}_{P_{C_\gamma}}[f'(C_\gamma)] > \mathbb{E}_{P_C}[f'(C)]$$

for all $P_C \ne P_{C_\gamma}$. Since $\mathbb{E}_{P_{\overline{C}}}[f'(\overline{C})] > \mathbb{E}_{P_C}[f'(C)]$ for all $P_C \ne P_{\overline{C}}$, and by Corollary A1, we have that $\mathbb{E}_{P_{C_\gamma}}[f'(C_\gamma)] = \mathbb{E}_{P_{\overline{C}}}[f'(\overline{C})]$, which completes the proof.

## D  PROOF FOR PROPOSITION 4

**Proposition A4 (Restated proposition 4 in main text)** *For* $\mathrm{MMD}(P_{\overline{C}}, P_{X'}, P_{Y'})$ *as defined in proposition 1,* $\widehat{C}_{\hat{\gamma}}$ *as defined in equation 7 and $\kappa$ such that $0 \le k(x,y) \le \kappa$ for all $x,y \in \mathcal{X}$. Then as for a sufficiently small $\epsilon$:*

$$P_{X',Y'}\left\{ |\mathrm{MMD}(P_{\overline{C}}, P_{X'}, P_{Y'}) - \widehat{\mathrm{MMD}}(\widehat{C}_{\hat{\gamma}}, X', Y')| > b_0 + \varepsilon \right\} \le 2\exp\left(\frac{-\varepsilon^2 n}{b_1}\right)$$

*for* $b_0 = 4(\kappa/n)^{1/2}(1+\epsilon)$ *and* $b_1 = 2\kappa\big((1-\epsilon)^3 + (1+\epsilon)(1+3\epsilon)^2\big)$

*Proof.* Consider the absolute difference term

$$|\mathrm{MMD}(P_{\overline{C}}, P_{X'}, P_{Y'}) - \widehat{\mathrm{MMD}}(\widehat{C}_{\hat{\gamma}}, X', Y')|$$

$$= \left| \sup_f \left[ (1-\epsilon)\mathbb{E}_{P_{X'}} f(X') - (1+\epsilon)\mathbb{E}_{P_{Y'}} f(Y') + 2\epsilon\mathbb{E}_{\overline{C}} f(\overline{C}) \right] \right.$$
$$\left. - \sup_f \left[ \frac{(1-\epsilon)}{n}\sum_i f(x_i') - \frac{(1+\epsilon)}{n}\sum_i f(y_i') + \frac{2\epsilon}{n}\sum_i f(\hat{c}_i^{\hat{\gamma}}) \right] \right|$$

$$\le \sup_f \left| (1-\epsilon)\mathbb{E}_{P_{X'}} f(X') - (1+\epsilon)\mathbb{E}_{P_{Y'}} f(Y') + 2\epsilon\mathbb{E}_{\overline{C}} f(\overline{C}) \right.$$
$$\left. - \frac{(1-\epsilon)}{n}\sum_i f(x_i') + \frac{(1+\epsilon)}{n}\sum_i f(y_i') - \frac{2\epsilon}{n}\sum_i f(\hat{c}_i^{\hat{\gamma}}) \right|$$

$$= \sup_f \left| (1-\epsilon)\mathbb{E}_{P_{X'}} f(X') - (1+\epsilon)\mathbb{E}_{P_{Y'}} f(Y') + 2\epsilon\mathbb{E}_{\overline{C}} f(\overline{C}) \right.$$
$$\left. - \frac{(1-\epsilon)}{n}\sum_i f(x_i') + \frac{(1+\epsilon)}{n}\sum_i f(y_i') - \frac{2\epsilon}{n}\sum_i \mathbb{1}\{f(y_i') \ge \hat{\gamma}\}f(y_i') \right|$$

$$:= \Delta_{\mathcal{D}}(P_{X'}, P_{Y'}, X', Y')$$

We will next attempt to bound the difference between $\Delta_{\mathcal{D}}(P_{X'}, P_{Y'}, X', Y')$ and its expectation by applying McDiarmid's inequality. To do so, we first need to verify that $\Delta_{\mathcal{D}}(P_{X'}, P_{Y'}, X', Y')$ satisfies the bounded difference property. We do so in two steps. In the first step, we consider the case where we replace one of the $X'$ samples. Specifically, we consider the data $\mathcal{D}_{\pi j}^{X'} = \{X'_{\pi j}, Y'\}$, where $X'_{\pi j} = \{x'_1, x'_2, \ldots, x'_{i-1}, x'_j, x'_{i+1}, \ldots x'_{(1-\epsilon)n}\}$. In that case, we have that:

$$|\Delta_{\mathcal{D}}(P_{X'}, P_{Y'}, X', Y') - \Delta_{\mathcal{D}_j^{X'}}(P_{X'}, P_{Y'}, X'_{\pi j}, Y')|$$

$$= \sup_f \left|(1-\epsilon)\mathbb{E}_{P_{X'}}f(X') - (1+\epsilon)\mathbb{E}_{P_{Y'}}f(Y') + 2\epsilon\mathbb{E}_{\overline{C}}f(\overline{C})\right.$$

$$- \frac{(1-\epsilon)}{n}\sum_i f(x'_i) + \frac{(1+\epsilon)}{n}\sum_i f(y'_i) - 2\epsilon\frac{1}{n}\sum_i \mathbb{1}\{f(y'_i) \geq \hat{\gamma}\}f(y'_i)\Big|$$

$$- \sup_f \left|(1-\epsilon)\mathbb{E}_{P_{X'}}f(X') - (1+\epsilon)\mathbb{E}_{P_{Y'}}f(Y') + 2\epsilon\mathbb{E}_{\overline{C}}f(\overline{C})\right.$$

$$- \frac{(1-\epsilon)}{n}\sum_i f(x'_i) + \frac{(1+\epsilon)}{n}\sum_i f(y'_i) - 2\epsilon\frac{1}{n}\sum_i \mathbb{1}\{f(y'_i) \geq \tilde{\gamma}\}f(y'_i) + \frac{1-\epsilon}{n}(f(x'_j) - f(x'_i))\Big|$$

$$\leq \sup_{f,\gamma} \left|(1-\epsilon)\mathbb{E}_{P_{X'}}f(X') - (1+\epsilon)\mathbb{E}_{P_{Y'}}f(Y') + 2\epsilon\mathbb{E}_{\overline{C}}f(\overline{C})\right.$$

$$- \frac{(1-\epsilon)}{n}\sum_i f(x'_i) + \frac{(1+\epsilon)}{n}\sum_i f(y'_i) - 2\epsilon\frac{1}{n}\sum_i \mathbb{1}\{f(y'_i) \geq \gamma\}f(y'_i)\Big|$$

$$- \sup_f \left|(1-\epsilon)\mathbb{E}_{P_{X'}}f(X') - (1+\epsilon)\mathbb{E}_{P_{Y'}}f(Y') + 2\epsilon\mathbb{E}_{\overline{C}}f(\overline{C})\right.$$

$$- \frac{(1-\epsilon)}{n}\sum_i f(x'_i) + \frac{(1+\epsilon)}{n}\sum_i f(y'_i) - 2\epsilon\frac{1}{n}\sum_i \mathbb{1}\{f(y'_i) \geq \gamma\}f(y'_i) + \frac{1-\epsilon}{n}(f(x'_j) - f(x'_i))\Big|$$

$$\leq \frac{1-\epsilon}{n}\sup_f \left|(f(x'_i) - f(x'_j))\right|$$

$$\leq \frac{1-\epsilon}{n}\left(\sup_f |(f(x'_i)| + \sup_f |f(x'_j))|\right)$$

$$\leq \frac{2(1-\epsilon)}{n}\sqrt{\kappa} \tag{13}$$

Second, we consider the case where we replace one of the $Y'$ samples. Specifically, we consider the data $\mathcal{D}_{\pi j}^{Y'} = \{X', Y'_{\pi j}\}$, where $Y'_{\pi j} = \{y'_1, y'_2, \ldots, y'_{i-1}, y'_j, y'_{i+1}, \ldots y'_{(1+\epsilon)n}\}$. In that case, we have that:

$$|\Delta_{\mathcal{D}}(P_{X'}, P_{Y'}, X', Y') - \Delta_{\mathcal{D}_j^{Y'}}(P_{X'}, P_{Y'}, X', Y'_{\pi j})|$$

$$\leq \sup_f \left|(1-\epsilon)\mathbb{E}_{P_{X'}}f(X') - (1+\epsilon)\mathbb{E}_{P_{Y'}}f(Y') + 2\epsilon\mathbb{E}_{\overline{C}}f(\overline{C})\right.$$

$$- \frac{(1-\epsilon)}{n}\sum_i f(x'_i) + \frac{(1+\epsilon)}{n}\sum_i f(y'_i) - 2\epsilon\frac{1}{n}\sum_i \mathbb{1}\{f(y'_i) \geq \hat{\gamma}\}f(y'_i)\Big|$$

$$- \sup_f \left|(1-\epsilon)\mathbb{E}_{P_{X'}}f(X') - (1+\epsilon)\mathbb{E}_{P_{Y'}}f(Y') + 2\epsilon\mathbb{E}_{\overline{C}}f(\overline{C})\right.$$

$$- \frac{(1-\epsilon)}{n}\sum_i f(x'_i) + \frac{(1+\epsilon)}{n}\sum_i f(y'_i) - 2\epsilon\frac{1}{n}\sum_i \mathbb{1}\{f(y'_i) \geq \tilde{\gamma}\}f(y'_i)$$

$$- \frac{1+\epsilon}{n}(f(y'_i) - f(y'_j)) + \frac{2\epsilon}{n}(\mathbb{1}\{f(y'_i) \geq \hat{\gamma}\}f(y'_i) - \mathbb{1}\{f(y'_j) \geq \tilde{\gamma}\}f(y'_j))\Big|$$

$$= \sup_f \left|- \frac{1+\epsilon}{n}(f(y'_i) - f(y'_j)) + \frac{2\epsilon}{n}(\mathbb{1}\{f(y'_i) \geq \hat{\gamma}\}f(y'_i) - \mathbb{1}\{f(y'_j) \geq \tilde{\gamma}\}f(y'_j))\right|$$

$$\leq \frac{1+\epsilon}{n}\sup_f \left|(f(y'_i) - f(y'_j))\right| + \frac{2\epsilon}{n}\sup_f \left|(\mathbb{1}\{f(y'_i) \geq \hat{\gamma}\}f(y'_i) - \mathbb{1}\{f(y'_j) \geq \tilde{\gamma}\}f(y'_j))\right|$$

$$\leq \frac{1+\epsilon}{n} \sup_f \left| (f(y_i') - f(y_j'))\right| + \frac{2\epsilon}{n} \sup_f |f(y_i') - f(y_j')|$$

$$\leq \frac{1+\epsilon}{n} \left( \sup_f |(f(y_i')| + \sup_f |f(y_j'))| \right) + \frac{2\epsilon}{n} \left( \sup_f |f(y_i')| + \sup_f |f(y_j')| \right)$$

$$\leq \frac{2(1+\epsilon)}{n} \sqrt{\kappa} + \frac{4\epsilon}{n} \sqrt{\kappa} = \frac{2\sqrt{\kappa}}{n}(1+3\epsilon) \tag{14}$$

Combining the results from equations 13 and 14, we get that we can apply McDiarmid with the following denominator:

$$(1-\epsilon)n\left(\frac{2(1-\epsilon)}{n}\sqrt{\kappa}\right)^2 + (1+\epsilon)n\left(\frac{2\sqrt{\kappa}}{n}(1+3\epsilon)\right)^2 = \frac{4\kappa}{n}\left((1-\epsilon)^3 + (1+\epsilon)(1+3\epsilon)^2\right),$$

to obtain

$$P_{X',Y'}\left\{ \Delta_{\mathcal{D}}(P_{X'}, P_{Y'}, X', Y') - \mathbb{E}_{X',Y'}\left[\Delta_{\mathcal{D}}(P_{X'}, P_{Y'}, X', Y')\right] > \varepsilon \right\} \tag{15}$$

$$\leq 2\exp\left( \frac{-\varepsilon^2 n}{2\kappa\left((1-\epsilon)^3 + (1+\epsilon)(1+3\epsilon)^2\right)} \right). \tag{16}$$

Next, we seek to control the expectation, $\mathbb{E}_{X',Y'}\left[\Delta_{\mathcal{D}}(P_{X'}, P_{Y'}, X', Y')\right]$. To do so we use symmetrization Van Der Vaart et al. [1996]. Let $X^\bullet$ and $Y^\bullet$ be i.i.d samples of sizes $(1-\epsilon)n$ and $(1+\epsilon)n$ respectively, we have that:

$$\mathbb{E}_{X',Y'}\left[\Delta_{\mathcal{D}}(P_{X'}, P_{Y'}, X', Y')\right]$$

$$= \mathbb{E}_{X',Y'} \sup_f \left| (1-\epsilon)\mathbb{E}_{P_{X'}} f(X') - \frac{1-\epsilon}{n}\sum_i f(x_i') - (1+\epsilon)\mathbb{E}_{P_{Y'}} f(Y') + \frac{1+\epsilon}{n}\sum_i f(y_i') \right.$$

$$\left. + 2\epsilon \mathbb{E}_{\overline{C}} f(\overline{C}) - \frac{2\epsilon}{n}\sum_i \mathbb{1}\{f(y_i') \geq \hat{\gamma}\} f(y_i') \right|$$

$$= \mathbb{E}_{X',Y'} \sup_f \left| (1-\epsilon)\mathbb{E}_{X^\bullet}\left(\frac{1}{n}\sum_i f(x_i^\bullet)\right) - \frac{1-\epsilon}{n}\sum_i f(x_i') \right.$$

$$- (1+\epsilon)\mathbb{E}_{Y^\bullet}\left(\frac{1}{n} f(y_i^\bullet)\right) + \frac{1+\epsilon}{n}\sum_i f(y_i')$$

$$\left. + 2\epsilon \mathbb{E}_{Y^\bullet}\left(\frac{1}{n}\mathbb{1}\{f(y_i^\bullet) \geq \gamma^\bullet\} f(y_i^\bullet)\right) - \frac{2\epsilon}{n}\sum_i \mathbb{1}\{f(y_i') \geq \hat{\gamma}\} f(y_i') \right|$$

$$\leq \mathbb{E}_{X',Y',X^\bullet,Y^\bullet} \sup_f \left| \frac{1-\epsilon}{n}\sum_i f(x_i^\bullet) - \frac{1-\epsilon}{n}\sum_i f(x_i') - \frac{1+\epsilon}{n}\sum_i f(y_i^\bullet) + \frac{1+\epsilon}{n}\sum_i f(y_i') \right.$$

$$\left. + \frac{2\epsilon}{n}\sum_i \mathbb{1}\{f(y_i^\bullet) \geq \gamma^\bullet\} f(y_i^\bullet) - \frac{2\epsilon}{n}\sum_i \mathbb{1}\{f(y_i') \geq \hat{\gamma}\} f(y_i') \right|$$

$$\leq \mathbb{E}_{X',Y',X^\bullet,Y^\bullet} \sup_{f,\gamma} \left| \frac{1-\epsilon}{n}\sum_i f(x_i^\bullet) - \frac{1-\epsilon}{n}\sum_i f(x_i') - \frac{1+\epsilon}{n}\sum_i f(y_i^\bullet) + \frac{1+\epsilon}{n}\sum_i f(y_i') \right.$$

$$\left. + \frac{2\epsilon}{n}\sum_i \mathbb{1}\{f(y_i^\bullet) \geq \gamma\} f(y_i^\bullet) - \frac{2\epsilon}{n}\sum_i \mathbb{1}\{f(y_i') \geq \gamma\} f(y_i') \right|$$

$$\leq \mathbb{E}_{X',Y',X^\bullet,Y^\bullet,\sigma',\sigma^\bullet} \sup_{f,\gamma} \left| \frac{1-\epsilon}{n}\sum_i \sigma_i'(f(x_i^\bullet) - f(x_i')) + \frac{1+\epsilon}{n}\sum_i \sigma_i^\bullet(f(y_i^\bullet) - f(y_i')) \right.$$

$$\left. + \frac{2\epsilon}{n}\sum_{y_i',y_i^\bullet \geq \gamma} \sigma_i^\bullet(f(y_i^\bullet) - f(y_i')) \right|$$

$$\leq \mathbb{E}_{X',X^\bullet,\sigma} \sup_{f,\gamma} \left| \frac{1-\epsilon}{n}\sum_i \sigma_i'(f(x_i^\bullet) - f(x_i')) \right| + \mathbb{E}_{Y',Y^\bullet,\sigma} \sup_{f,\gamma} \left| \frac{1+\epsilon}{n}\sum_i \sigma_i^\bullet(f(y_i^\bullet) - f(y_i')) \right|$$

$$+ \mathbb{E}_{Y',Y^\bullet,\sigma} \sup_{f,\gamma} \left| \frac{2\epsilon}{n}\sum_{y_i',y_i^\bullet \geq \gamma} \sigma_i^\bullet(f(y_i^\bullet) - f(y_i')) \right|$$

$$\leq 2[(1-\epsilon)\mathcal{R}_n(\mathcal{F}, X') + (1+\epsilon)\mathcal{R}_n(\mathcal{F}, Y') + 2\epsilon\mathcal{R}_n(\mathcal{F}, Y')]$$

$$\leq 2[(1-\epsilon)\left(\frac{\kappa}{n}\right)^{1/2} + (1+3\epsilon)\left(\frac{\kappa}{n}\right)^{1/2}] = 4\left(\frac{\kappa}{n}\right)^{1/2}(1+\epsilon)$$

Substituting $4\left(\frac{\kappa}{n}\right)^{1/2}(1+\epsilon)$ in equation 15 yields the desired result.

# E   ADDITIONAL RESULTS FROM THE NONRANDOM CONTAMINATION SETTING

We show results presenting the typical estimate of the MMD assuming no contamination. We also reproduce the main results in sections 4 in the MIMIC setting. We additionally include the same experiment as in table 1 for $N = 2000$.

Figure 4 illustrates the that the typical estimate of the MMD (equation 1) is unreliable, especially as $\epsilon$ increases. It also demonstrates the upper and lower bounds of S-SD as simulated epsilon contamination increases; S-SD bounds contain the true value of the MMD at all values of $\epsilon$.

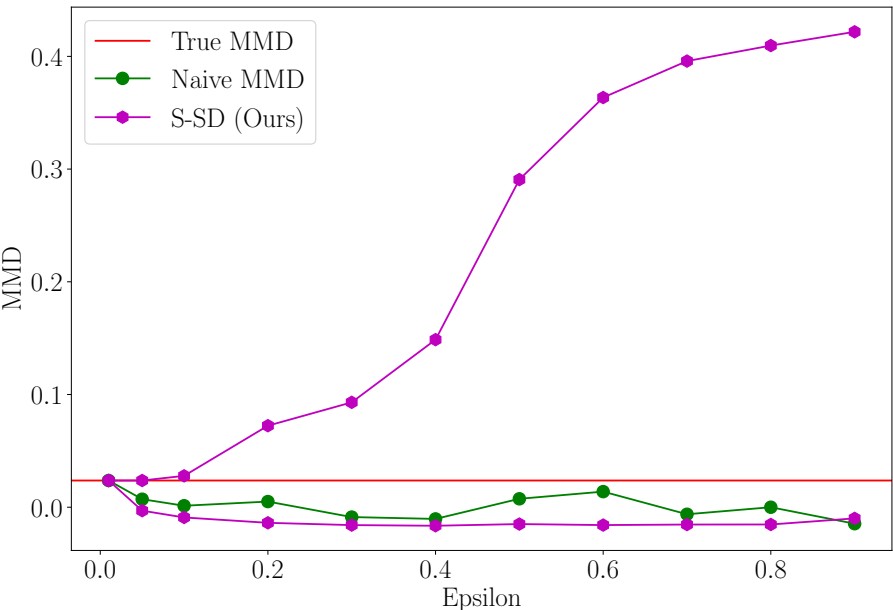

Figure 4: An illustration of how the typical estimate of the MMD is unreliable especially as $\epsilon$ increases. In addition, this result matches our intuition from the Bootstrap method; as $\epsilon$ increases, the two groups become increasingly mixed and more similar, and the MMD approaches 0.

Table 3 shows the same results as those presented in table 1 with $N = 2000$ instead of $N = 100$. The results show that, as seen in figure 1 (top), the performance of QNO and S-QNO improves as sample size increases, while S-SD continues to have tight and informative bounds.

Table 3: MIW and FCR for MIMIC and FOREST at $\epsilon = 0.2$.

| Approach | MIMIC ($n = 2000, d = 2$) | | FOREST ($n = 2000, d = 54$) | |
| | FCR | MIW | FCR | MIW |
|---|---|---|---|---|
| S-SD (Ours) | $0.0 \pm (0.0)$ | $0.251 \pm (0.008)$ | $0.0 \pm (0.0)$ | $0.128 \pm (0.007)$ |
| S-QNO | $0.0 \pm (0.0)$ | $0.25 \pm (0.006)$ | $0.0 \pm (0.0)$ | $0.134 \pm (0.007)$ |
| QNO | $0.0 \pm (0.0)$ | $0.227 \pm (0.006)$ | $0.32 \pm (0.066)$ | $0.107 \pm (0.01)$ |
| SD | $0.02 \pm (0.02)$ | $0.23 \pm (0.009)$ | $0.46 \pm (0.07)$ | $0.087 \pm (0.009)$ |
| SM | $0.02 \pm (0.02)$ | $0.217 \pm (0.008)$ | $0.46 \pm (0.07)$ | $0.081 \pm (0.008)$ |
| Bootstrap | $0.3 \pm (0.065)$ | $0.091 \pm (0.004)$ | $0.46 \pm (0.07)$ | $0.042 \pm (0.003)$ |

## E.1 ADDITIONAL RESULTS USING MIMIC DATA

Figures 5 and 6 are similar to figure 1 in the main text, but instead of performing the analysis on the FOREST data, we perform the analysis on the MIMIC data. The results are largely consistent with the analysis in the main text: our approach outperforms others in that it gives the lowest FCR for every sample size and every value of $\epsilon$.

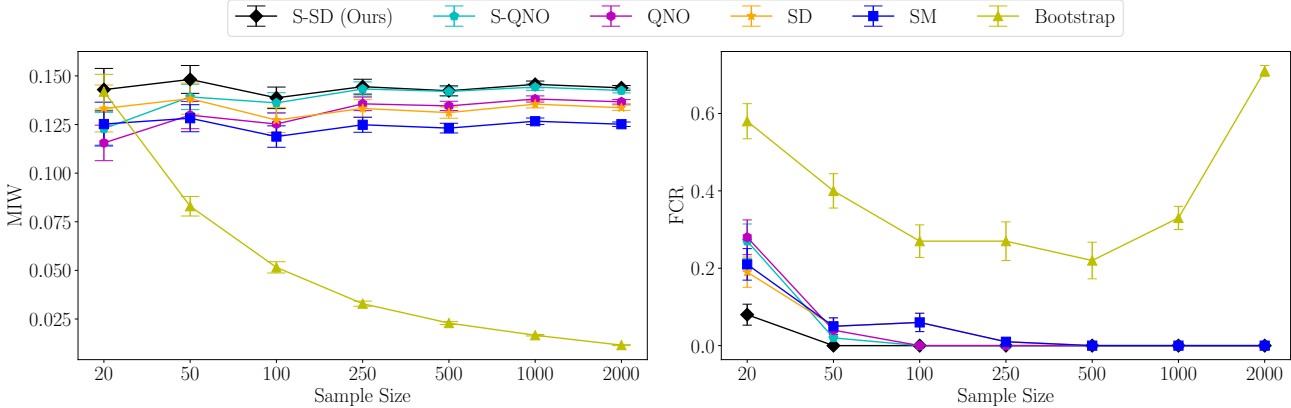

Figure 5: The same experiment as in figure 1 (top), but run in the MIMIC ($n = 100, d = 2$) setting.

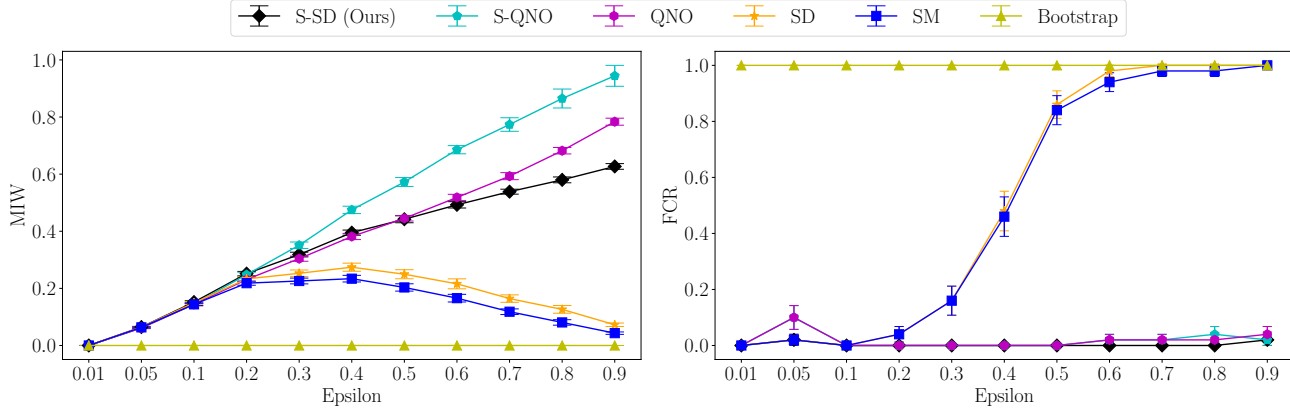

Figure 6: The same experiment as in figure 1 (bottom), but run in the MIMIC ($n = 100, d = 2$) setting.

## E.2 ADDITIONAL RESULTS USING BIO DATA

Figure 7 is similar to figure 1 (bottom) in the main text, but instead of performing the analysis on the FOREST data, we perform the analysis on the BIO data. We note that due to the limited sample size of the BIO data, we are unable to create figure 1 (top) for the BIO data. S-SD gives the lowest FCR for every value of $\epsilon$. As in 1, QNO and S-QNO have a irreducible dependence on the dimension size of the data. QNO fails to contain the value of the true MMD at all $\epsilon \geq 0.01$. S-QNO performs poorly until larger values of epsilon, where the step approximation becomes effective; this is because at small sample sizes, the set of corrupted samples is small, and the approximation cannot be divided into many steps.

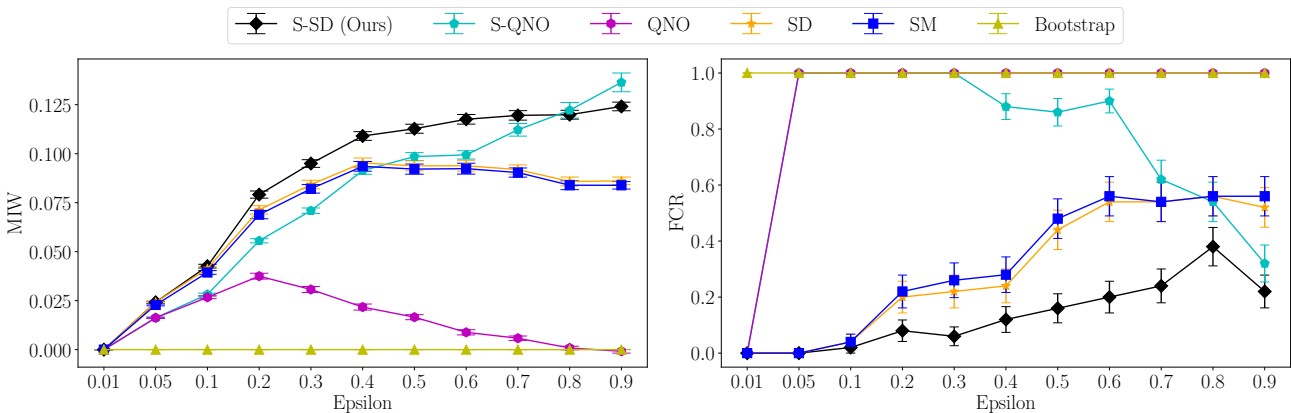

Figure 7: The same experiment as in figure 1 (bottom), but run in the BIO ($n = 72, d = 7128$) setting.

## E.3 STEP SIZE SENSITIVITY

Table 4 shows that similar to S-SD, S-QNO gives bound estimates with FCR of zero even for a few number of steps. Conclusions from the main text regarding setting the step size for S-SD hold for S-QNO as well.

|  | S-QNO | |
|---|---|---|
| Number of Steps | FCR | MIW |
| 2 | $0.05 \pm (0.023)$ | $0.067 \pm (0.001)$ |
| 3 | $0.0 \pm (0.0)$ | $0.066 \pm (0.001)$ |
| 5 | $0.0 \pm (0.0)$ | $0.075 \pm (0.0)$ |
| 10 | $0.0 \pm (0.0)$ | $0.091 \pm (0.001)$ |
| 20 | $0.0 \pm (0.0)$ | $0.087 \pm (0.001)$ |
| 50 | $0.0 \pm (0.0)$ | $0.091 \pm (0.001)$ |

Table 4: Varying number of steps for S-QNO in FOREST ($n = 2000, d = 54$) with $\epsilon = 0.2$. Standard errors (shown in parentheses) represent the SE for the FCR and MIW for each method over 100 trials. In each trial, we sample 2000 data points without replacement and simulate $\epsilon$-contamination, and then compute the bounds for S-QNO at each number of steps on the same sample.

## F EXPERIMENTAL RESULTS FROM THE RANDOM CONTAMINATION SETTING

We present the same experiments as in table 1 and figure 1 on FOREST ($n = 100, d = 54$) when the set of contaminations $C^*$ is a random sample of $X$ of size $\lfloor \epsilon n \rfloor$, rather than the $\lfloor \epsilon n \rfloor$ samples in $X$ with the largest witness function values as described in section 5. The results in table **??** and figure 8 are consistent with the results in the main text and show that for all $\epsilon$, S-SD gives the most credible bounds with the tightest MIW. Figure 9 shows that FCR and MIW decrease for S-SD, S-QNO, and QNO as sample size increases in FOREST.

|  | MIMIC ($n = 100, d = 2$) | | FOREST ($n = 100, d = 54$) | | BIO ($n = 72, d = 7128$) | |
|---|---|---|---|---|---|---|
| Approach | FCR | MIW | FCR | MIW | FCR | MIW |
| S-SD (Ours) | $0.0 \pm (0.0)$ | $0.258 \pm (0.002)$ | $0.0 \pm (0.0)$ | $0.107 \pm (0.002)$ | $0.07 \pm (0.026)$ | $0.08 \pm (0.001)$ |
| S-QNO | $0.0 \pm (0.0)$ | $0.258 \pm (0.002)$ | $0.0 \pm (0.0)$ | $0.114 \pm (0.002)$ | $1.0 \pm (0.0)$ | $0.056 \pm (0.001)$ |
| QNO | $0.0 \pm (0.0)$ | $0.247 \pm (0.002)$ | $0.4 \pm (0.069)$ | $0.051 \pm (0.002)$ | $1.0 \pm (0.0)$ | $0.038 \pm (0.001)$ |
| SD | $0.0 \pm (0.0)$ | $0.24 \pm (0.002)$ | $0.92 \pm (0.038)$ | $0.064 \pm (0.003)$ | $0.15 \pm (0.036)$ | $0.074 \pm (0.001)$ |
| SM | $0.0 \pm (0.0)$ | $0.225 \pm (0.002)$ | $0.92 \pm (0.038)$ | $0.06 \pm (0.003)$ | $0.42 \pm (0.049)$ | $0.05 \pm (0.002)$ |
| Bootstrap | $1.0 \pm (0.0)$ | $0.02 \pm (0.0)$ | $0.6 \pm (0.069)$ | $0.005 \pm (0.0)$ | $0.85 \pm (0.036)$ | $0.036 \pm (0.001)$ |

Table 5: 100 Samples random contaminations

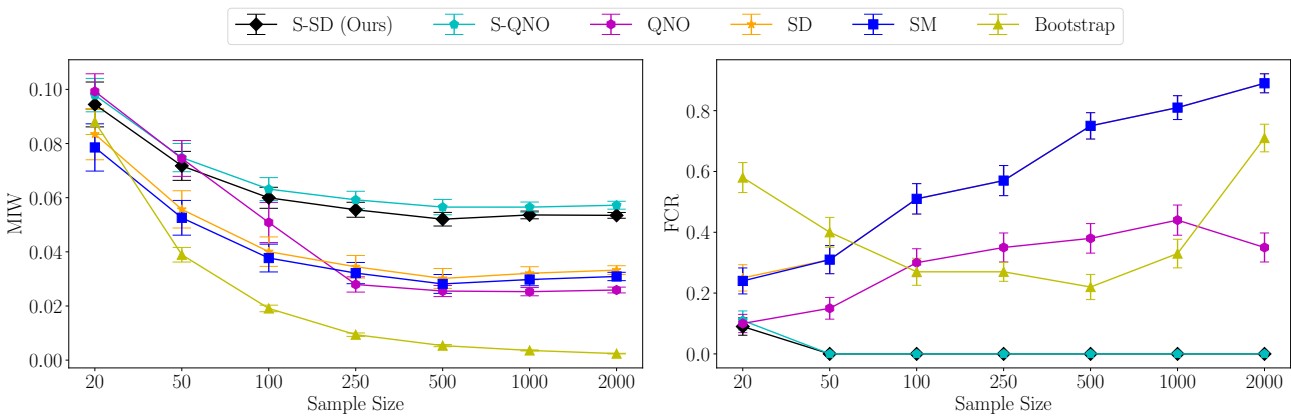

Figure 8: The MIW and FCR for each approach is shown as the sample size increases when $\epsilon = 0.2$ in FOREST ($n = 100, d = 54$). Bars indicate the SE of the FCR and MIW across all trials.

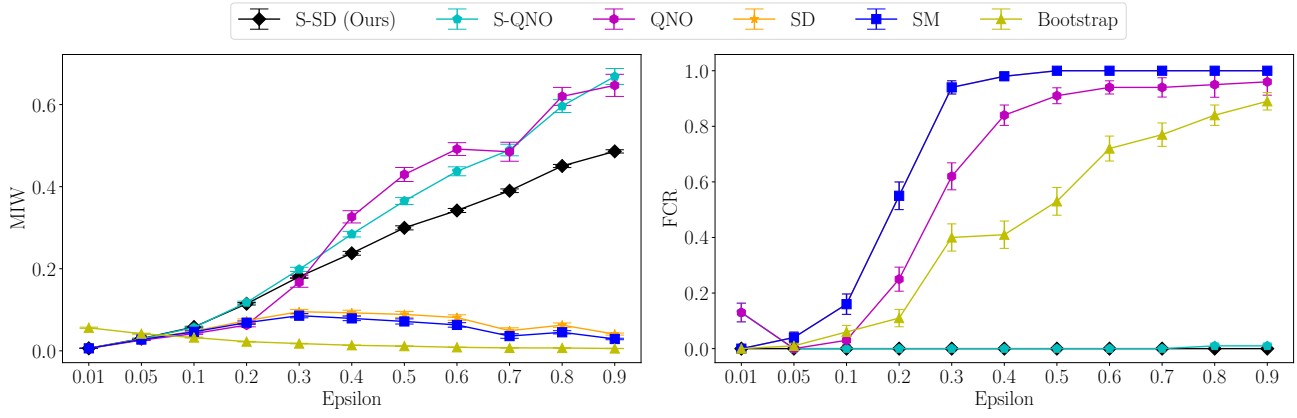

Figure 9: The MIW and FCR for each approach is shown as the intensity of random $\epsilon$-contamination varies from $\epsilon = 0.01$ to $\epsilon = 0.9$ in FOREST ($n = 100, d = 54$). Bars indicate the SE of the FCR and MIW across all trials.

# G   ADDITIONAL SENSITIVITY ANALYSIS RESULTS

Figure 10 shows $\tilde{\epsilon}$, the incorrect value of $\epsilon$, on the $x$-axis and its corresponding mean interval width (MIW) on the $y$-axis. The results show that the mean interval width increases – as expected – for our two main approaches (S-SD and S-QNO) but not for the other approaches. The latter give overly conservative estimates with high FCR at high values of $\tilde{\epsilon}$.

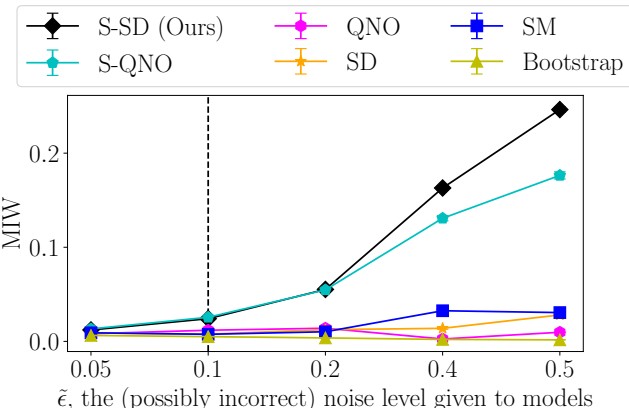

Figure 10: Sensitivity to incorrect values of $\epsilon$. True value of $\epsilon$ is 0.1. Models are given $\tilde{\epsilon}$ values shown on the $x$-axis, mean interval width (MIW) on the $y$-axis