# OpenReview forum: "Partial identification of the maximum mean discrepancy with mismeasured data"
_auai.org/UAI/2024/Conference — UAI 2024 poster_

### Official Review · Reviewer_uY2t · 2024-03-13

**Q2-1 Originality-Novelty:** 3
**Q2-2 Correctness-Technical Quality:** 1
**Q2-5 Clarity Of Writing:** 2

**Q10 Ethical Concerns:**

No.

**Q1 Summary And Contributions:**

This paper analyses the maximum mean discrepancy (MMD) between two samples but in a scenario of imprecision in the data. Specifically, the authors assume that a fixed percentage of data in a sample is contaminated and belongs to the other sample. In this situation, they adapt the MMD for comparing two samples, propose algorithms for its computation/estimation, apply them to some datasets and perform a sensitivity analysis.

**Q2-3 Extent To Which Claims Are Supported By Evidence:**

3: Good: the main claims are supported by convincing evidence (in the form of adequate experimental evaluation, proofs, (pseudo-)code, references, assumptions).

**Q2-4 Reproducibility:**

3: Good: key resources (e.g. proofs, code, data) are available and key details (e.g. proofs, experimental setup) are sufficiently well-described for competent researchers to confidently reproduce the main results.

**Q3 Main Strengths:**

Originality-Novelty: The idea of incorporating contaminated data is, to the best of my knowledge, a new approach in the literature. Moreover, this study is properly justified in the introduction.

**Q4 Main Weakness:**

-Correctness-Technical Quality: Parts of the paper are not properly formalised. The mathematical notation is not adequate and should be checked.
-Clarity Of Writing: Some parts of the paper and some assumptions are not properly justified.
-The term epsilon-contamination is used in the literature for a different thing.
-I miss an important reference in the bibliography that, I guess, comes from the same authors. There is a clear overlap between them.

**Q5 Detailed Comments To The Authors:**

Main comments: The authors propose a very interesting study of the comparison of samples when part of the data is contaminated. Nevertheless, there are some parts of the paper that should be better formalised, and some approaches should be better justified. Moreover, the authors should take into account the literature regarding robust statistics and probability modelling where the \epsilon-contamination model is used. As well, there is an important problem of overlap with a previous contribution that has been published in Arxiv. Specifically, the main issues are the following.

1.The idea of assuming that the data are contaminated is very interesting. However, the approach followed by the authors is rather strange: they assume that only one sample is contaminated, and by “contamination” they mean that that data come from the other sample. Here, the first problem appears:
ISSUE 1: In page 2, the authors assume that both sample sizes coincide (n), but this is not completely true: just after this comment, they say that n+m data come from X while n-m come from Y. Hence, the sample size cannot be the same.

2.Even though the contaminated data in the sample of Y comes from X, the authors say that they do not make any assumption on the distribution of what they call P_C*. Here, the second problem appears:
ISSUE 2: If those contaminated data called C* come from X, these should follow the distribution P_X. Assuming that “we do not make any additional assumptions about P_C*” makes no sense, because those data come from X so you do know their distribution.
This point should be clarified.

3.ISSUE 3: Parts of the paper are not properly formalised. The authors use P_X(X) to refer, I guess, the probability distribution of X. This makes no sense, it should be P_X. Similarly, it makes no sense to say (end of the first column in page 2) that “X=\{x_i\}_i^n”. X is a random variable, not a sample. Also, P_A(A) and E_{P_A}[A] make no sense, it should be P_A and E_{P_A}. There are several issues that should be fixed.

4.The term “\epsilon-contamination” is used in the literature (see for example papers or books by Huber, such as “Robust statistics”) to refer to the set of probability distributions \{\epsilon P+(1-\ epsilon)P_0 \forall probability measure P\}. This means that it is assumed that with probability 1-\epsilon, P_0 is the correct probability measure, while with probability \epsilon (that is assumed to be small enough) the correct probability measure is completely unknown and it could be any other probability measure.
ISSUE 4: please, check the literature to see connections with what Huber refers to epsilon-contamination and do the needed adaptations.

5.In page 4, paragraph “Approximation strategy for a sufficiently small epsilon”, the authors assume that they can remove of the terms in Eq.3 can be removed, and later they use stochastic dominance. What is removed is the only term in Eq.3 that takes into account the dependence between the samples from X and Y. Stochastic dominance only uses the marginal distributions of the random variables, then ignoring the possible dependence between them.
ISSUE 5: Under the assumption of this paragraph, it makes sense to use stochastic dominance. However, in general my feeling is that using stochastic dominance to compare the samples losses a lot of information about the dependence between the samples. Other stochastic orders such as precedence order, statistical preference or stochastic dominance in differences could be used here to consider the dependence.

Overall evaluation: taking into account the main issues aforementioned, I strongly believe that the manuscript should be thoroughly review of the manuscript to consider the previous comments. So far, my evaluation of the manuscript is between “borderline accept” and “borderline reject”.

Minor comments:

-P1, C1, L9: “…upper and lower bound…” -> “…upper and lower boundS…”.

-P1, C1, L16: “uncertain” is not the appropriate work here because you are assuming that \epsilon is fixed, not random. Do you mean “unknown”, “partially (un)known”, “imprecise”,…?

-P1, C2: “…topological spaces…”: do you mean possibility spaces? Or are you assuming some topological condition on the spaces (in such case, detail them)?

-Def.1: What is \mathcal{Z}?

-P3, after Eq.2: “categorizes” -> “characterizes”?

-P3, after Prop.1: what do you mean by “…C* can take on any values in \mathcal{Y}’…”?

-P3, C2., before Eq.4: what is \psi? What do you mean by C\in Y’? I refer to the main issue 3.

-P4, Eq.5: remove final dot.

**Q9 Complying With Reviewing Instructions:**

Yes

---

> ### Author Rebuttal · Authors · 2024-04-05
>
> We thank the reviewer for their time and diligence in reviewing our work. We are very encouraged that the reviewer recognizes the novelty of our work and appreciates that our work is properly justified, and fits well with the UAI themes. We respond to their questions below.
>
> 1. Issue #6 - Regarding overlap with an arxiv paper: We thank the reviewer for their diligent investigation of related work. We also wish to stress what the Program and Area Chairs have already communicated with the reviewer: that UAI policy permits submitting work that has been previously uploaded to arxiv even if the two versions are identical. We cannot further comment on this issue to preserve our anonymity.
>
> 2. Issue #1 - Sample sizes: we wish to clarify that we assume that the _uncontaminated_ samples are of size $n$ each. As described in the preliminaries section, the process of contamination removes a subsample of size $m= \epsilon n$ from X and groups it with Y. This gives us the contaminated samples X’ (of size $n-m$ since $m$ contaminated samples are erroneously excluded) and Y’ (of size $n+m$ since $m$ contaminated samples are erroneously included). We note in passing that this assumption is not necessary for our work: we make it out of notational convenience.
>
> 3. Issue #2 - The reviewer asks us to use $P_{X}$ to denote the distribution of $P_{C^*}$, which would imply that the contaminated sample $C^* \sim P_X$. We appreciate the reviewer’s efforts to help simplify our notation, however, the suggested notation would incorrectly imply that we are assuming that $C^*$ is sampled at random from $P_X$, which is a strong assumption that is likely violated in practice. Instead, we do not rely on this assumption, and let $P_X$ possibly be a mixture of $P_{X’}$ and some other $P_{C^*}$.
>
> 4. Issue #3 - Notational comments: We thank the reviewer for their remarks. We believe that slightly modifying our notation from $P_X(X)$ to $P\_\boldsymbol{X}(X)$ might clear the confusion here, where $P\_\boldsymbol{X}(X)= P(\boldsymbol{X} = X)$. As for $X$, we define $X$ to be a single draw (i.e., a sample) from the distribution $P\_\boldsymbol{X}$ in the preliminaries section. We are happy to address any other notational issues that the reviewer believes should be fixed if they can share what these issues are.
>
> 5. Issue #4 - Relation to Huber: The reviewer is correct to draw an analogy between this work and Huber’s pioneering work in this area. We wish to clarify that our nomenclature is intentional since our work builds upon the classical work of Horowitz and Manski (as we state in the last paragraph of the first column on page 4), which in turn builds upon Huber’s pioneering work. Specifically, Horowitz and Manski adopt this nomenclature but extend Huber’s work to the setting where some constraints are placed on the unknown probability measure (see the last paragraph of section 2.2 in the Horowitz and Manksi 1995 citation in our work for a more detailed explanation of the connection between the work we’re building upon and Huber’s robust statistics). We follow Horowitz and Manski in adopting this nomenclature since they are the closest to our work in terms of classical literature.
>
> 6. Issue #5 - Stochastic dominance: We wish to clarify that our work leverages stochastic dominance as defined with respect to the distribution over the witness function evaluated on all the data (without removing any data points). The dependence between variables is captured through the kernel function in the definition of the witness function. We also wish to stress that the term that is “removed” is multiplied by $\epsilon$. Under the assumptions of the paragraph that the reviewer mentions, $\epsilon$ (and hence that whole term) approaches zero so there is no loss of information.
>
> 7. Minor issues: We thank the reviewer for catching those typos, we are glad to correct them in the final version. We will also clarify the following
> a) $\mathcal{Z}$ is the support of $P_Z$, an arbitrary distribution
> b) Values of $C^*$: we mean that for a sample $Y’$, we have that $C^* \in Y’$ (recall that $Y’ = \\{ y’_i \\}_i^{m+n}$).
> c) $\psi$ is defined on the second line of equation 3. We state it here for completeness
> $$\psi(C, X', Y')=\frac{(1 - \epsilon)}{n}\sum_i \sum_j k(x'_i, c_j)-\frac{(1+\epsilon)}{n} \sum_i \sum_j k(y'_i, c_j)+\frac{\epsilon}{n} \sum_i \sum\_{j \not= i} k(c_i, c_j)$$

---

### Official Review · Reviewer_Qg3d · 2024-03-20

**Q2-1 Originality-Novelty:** 3
**Q2-2 Correctness-Technical Quality:** 3
**Q2-5 Clarity Of Writing:** 4

**Q10 Ethical Concerns:**

No.

**Q1 Summary And Contributions:**

This is an interesting paper on a very well-defined question: how to bound maximum mean discrepancy between (distributions of) X and Y when some (up to epsilon) X are misclassified as Y in the data. This gives rise to contamination neighborhoods and connects to “classical” statistics/econometrics work from Wasserman and Kadane (JASA) to Horowitz and Manski (Econometrica) and more. The motivation for the specific setting is that MMD is frequently used in the machine learning and algorithmic fairness literatures.

The problem is computationally involved and the contribution lies in practically solving it as well as giving guarantees for how quickly empirical bounds converge to population bounds.

**Q2-3 Extent To Which Claims Are Supported By Evidence:**

4: Excellent: all claims are supported by very convincing evidence (in the form of comprehensive experimental evaluation, rigorous mathematical proofs, detailed (pseudo-)code, precise references, well-motivated and realistic assumptions) and the authors deliver what they promise.

**Q2-4 Reproducibility:**

4: Excellent: key resources (e.g. proofs, code, data) are available and key details (e.g. proof sketches, experimental setup) are comprehensively described for competent researchers to confidently and easily reproduce the main results.

**Q3 Main Strengths:**

I do not have much to say on this one. The question strikes me as well-defined and the approach is convincing.

**Q4 Main Weakness:**

The paper is "safe" partly because it is somewhat incremental. The question is relatively obvious to ask if one knows kallus et al or a large literature in econometrics. I do not see this as a big issue.

**Q5 Detailed Comments To The Authors:**

I do not have major comments. Smaller ones:

•	Are the convergence rates optimal?

•	I have no intuition why the bootstrap approach you describe would be expected to work at all? (And it seemingly doesn’t, so no contradiction there. Just wondering why it is being considered.)

•	It’s tangential to the paper but I wondered about statistical inference and testing. Inference under partial identification is an ongoing research literature. Molinari (Handbook of Econometrics) is a recent overview. I am not aware of an approach that directly applies to the present setting. As bounds are defined as values of constrained optimization problems, basic ideas in Kaido, Molinari, Stoye (Econometrics 2019) might be relevant (i.e., solve an appropriately relaxed problem) but they assume a fully parametric setting. As a last straw, might subsampling (aka m<n-bootstrap) work? If it is not feasible to solve the problem many times, there are of course bootstrap approaches that avoid that but showing validity might be harder. However, these are just my ruminations, not a request to actually do any of this in the confines of the current paper.

**Q9 Complying With Reviewing Instructions:**

Yes

---

> ### Author Rebuttal · Authors · 2024-04-05
>
> We are thankful for the reviewer’s thoughtful insight. We are particularly thrilled that the reviewer recognizes our attempts to connect to and extend classical results in the econometrics and statistics literature.
>
> 1. The reviewer raises an interesting question about the optimality of our rates. There are multiple notions of optimality that could be considered here. As proposition 1 implies, the population upper and lower bounds are “optimal” in the sense that they are sharp. As far as finite sample optimality, we do not have such results, and can only show that our approach has more favorable properties compared to other approaches. We are not aware of any minimax optimality guarantees even for MMD estimates in the uncontaminated setting.
>
> 2. Regarding the bootstrap approach: We agree with the reviewer that the bootstrap approach is a weak baseline. We considered it to show that the “usual” methods for estimating uncertainty in the MMD are unsatisfactory here.
>
> 3. We appreciate the reviewer’s insights about related results in the econometrics literature. We are happy to include these for an extended and more comprehensive literature review in the final manuscript.

---

### Official Review · Reviewer_vA7V · 2024-03-21

**Q2-1 Originality-Novelty:** 2
**Q2-2 Correctness-Technical Quality:** 3
**Q2-5 Clarity Of Writing:** 3

**Q1 Summary And Contributions:**

The paper is concerned with estimating the maximum mean discrepancy (MMD) between two distributions  under bounded contamination, i.e. samples from one of the two distributions are erroneously grouped with the other. It gives upper and lower bounds on the MMD identifiable as a function of observed data and known degree of contamination. It also performs a sensitivity analysis of the bounds wrt incorrect values of degree of contamination.

**Q2-3 Extent To Which Claims Are Supported By Evidence:**

3: Good: the main claims are supported by convincing evidence (in the form of adequate experimental evaluation, proofs, (pseudo-)code, references, assumptions).

**Q2-4 Reproducibility:**

3: Good: key resources (e.g. proofs, code, data) are available and key details (e.g. proofs, experimental setup) are sufficiently well-described for competent researchers to confidently reproduce the main results.

**Q3 Main Strengths:**

Theory is quite cogent and clear. Estimation approach seems sophisticated. Empirical results are promising.

**Q4 Main Weakness:**

The contaminated discrepancy estimation problem seems somewhat niche. It’s unclear to me how restrictive the assumption that only one of the two variables is incorrectly grouped is.  Would be nice to have some examples of when you get a plausible estimate of epsilon, but I suppose the point of the analysis is to show how robust discrepancy judgements are to contamination.

**Q5 Detailed Comments To The Authors:**

Could you say something about the case when both variables can be incorrectly grouped? Is this a trivial extension, or a major undertaking?

I do not understand how you get the expressions for P_{Y’}(Y’) and P_{X’}(X’). I get something slightly different. Can you show your calculations?

**Q9 Complying With Reviewing Instructions:**

Yes

---

> ### Author Rebuttal · Authors · 2024-04-05
>
> We thank the reviewer for their insightful feedback! We are very pleased to know that the reviewer found our theory cogent and clear, our estimation approach sophisticated and our empirical results promising. We address the reviewer’s comments below
>
> 1. **Examples of when you can get a plausible estimate of epsilon:** **_First_**, the example we gave in the paper: silent myocardial infarction cases. Estimates of the number of these undiagnosed cases are well documented in the clinical literature, and are typically obtained from small clinical trials where the health of the participants is monitored independently of whether or not they express symptoms (e.g., see [1], table 2 for rates among the diabetic population). **_Second_**, in Medicare insurance claims data, the race variable is imputed for some of the beneficiaries. When an individual qualifies for Medicare through age or disability, their true race is obtained. By contrast, if an individual qualifies by being married to an enrollee, their race is assumed to be the same as their spouse’s race [2]. If we are trying to estimate difference in billed procedures between Black and White Medicare beneficiaries, we will end up in a contaminated setting like the one described in this paper. In this setting, a conservative estimate of $\epsilon$ can be the proportion of medicare beneficiaries who qualified through their spouse.  We will add this example to the final version of the paper.
>
> 2. The reviewer raises an interesting question: **what happens if both variables are contaminated?** We will briefly comment on the question, but as we mention in the conclusion, other measurement error mechanisms fall in the scope of future work. In this case, our solution extends trivially only if the two contaminated variables are independent of each other. If they are not, an iterative variation of our approach where we find some solution for the contamination in variable $X’$, and then based on that find the solution for the contamination in variable $Y’$, and then refine our solution to $X’$, and so forth until some convergence criteria is met is more appropriate. We will add this to the conclusion section in the final version.
>
> 3. We are happy to provide our derivation. Recall that $X,Y$ are the uncontaminated samples with length $n$ each. Let $G \in \\{0, 1\\}^n$ take a value of 1 if the corresponding value in $X$ is correctly included in $X'$ and zero otherwise. We have that:
> $$P_X(X) = P(X| G=1) P(G=1) + P(X| G=0) P(G=0) = P\_{X'}(X') (1 - \epsilon) + P\_{C^*}(C^*) \epsilon$$
> Solving for $P\_{X'}(X')$:
> $$P\_{X'}(X') = P_X(X) (\frac{1}{1- \epsilon}) - P\_{C^*}(C^*) (\frac{\epsilon}{1- \epsilon}) = P_X(X) (\frac{1}{1- \epsilon}) -  P\_{C^*}(C^*) \tilde{\alpha}$$
> $$= P_X(X) (\frac{1 + \epsilon - \epsilon}{1- \epsilon}) -  P\_{C^*}(C^*) \tilde{\alpha} = P_X(X) (\frac{\epsilon}{1- \epsilon} + \frac{1-\epsilon}{1- \epsilon}) -  P\_{C^*}(C^*) \tilde{\alpha} = P_X(X) (1 + \tilde{\alpha}) - P\_{C^*}(C^*) \tilde{\alpha}  $$
>
> The derivation for $Y'$ is more straight forward. Recall that $Y'$ is of size $n(1+\epsilon)$. Let $K \in \\{0, 1\\}^{n(1+\epsilon)}$ take a value of 1 if the corresponding value in $Y'$ truly belongs in $Y$ and zero otherwise. Note that $P(K=1) = \frac{n}{n(1+\epsilon)} = \frac{1}{(1+\epsilon)}$
>
> $$P_{Y}(Y') = P(Y'|K=1)P(K=1) + P(Y'|K=0)P(K=0) = P_Y(Y)(\frac{1}{1+\epsilon}) + P\_{C^*}(C^*)(\frac{\epsilon}{1+\epsilon})$$
> $$= P_Y(Y)(\frac{1 + \epsilon - \epsilon}{1+\epsilon}) + P\_{C^*}(C^*)\alpha =  P_Y(Y)(\frac{1 + \epsilon}{1+\epsilon}  - \frac{ \epsilon }{1+\epsilon}) + P\_{C^*}(C^*)\alpha = P_Y(Y)(1- \alpha) + P\_{C^*}(C^*)\alpha$$
>
> [1]  Scirica, Benjamin M. "Prevalence, incidence, and implications of silent myocardial infarctions in patients with diabetes mellitus." Circulation 127.9 (2013): 965-967.
>
> [2] Jarrín, Olga F., et al. "Validity of race and ethnicity codes in Medicare administrative data compared with gold-standard self-reported race collected during routine home health care visits." Medical care 58.1 (2020): e1-e8.

---

### Official Review · Reviewer_juJM · 2024-03-22

**Q2-1 Originality-Novelty:** 3
**Q2-2 Correctness-Technical Quality:** 3
**Q2-5 Clarity Of Writing:** 1

**Q1 Summary And Contributions:**

This paper proposes sharp and tight bounds for the true MMD with measurement errors.

**Q2-3 Extent To Which Claims Are Supported By Evidence:**

2: Fair: the main claims are somewhat supported by evidence (but the experimental evaluation may be weak, or does not match entirely with the claims, important baselines may be missing, proofs contain important ideas but lack rigor, algorithmic details are only discussed superficially, references are imprecise, assumptions are not sufficiently motivated or explicated, etc.).

**Q2-4 Reproducibility:**

2: Fair: key resources (e.g. proofs, code, data) are unavailable but key details (e.g. proof sketches, experimental setup) are sufficiently well-described for an expert to confidently reproduce the main results.

**Q3 Main Strengths:**

Interesting problem, which would be significant if it’s properly solved.

**Q4 Main Weakness:**

1. The problem setting is ambiguously defined. Specifically, it's hard for me to understand what's the meaning of "two variables are grouped". Without clear mental picture of the problem setting, any further results such as Proposition 1 is hard to understand.
2. Proposition 1 looks trivial. See the detailed comments.
3. A key quantity $\psi$ seems undefined, so it’s hard to assess the paper’s quality.
4. Technical presentations are made without further details, examples or explanations.

**Q5 Detailed Comments To The Authors:**

1. I don’t understand the meaning of Equation (1). Specifically, I am not sure$\operatorname{MMD}(P_C,P_{X'},P_{Y'})$ is a valid notation, given that $P_C$ is a distribution, not the function space (RKHS).

2. In Section 2, what does it mean for two variables to be grouped? I believe an explicit example of the problem setting is needed. The explanation of the problem setting is unclear without any examples, making the paper difficult to understand.

3. Proposition 1 appears very obvious. Essentially, Proposition 1 states that for any function $f(x,y,z)$, where $\mathcal{X}$ is a collection of all possible $x$, the inequality $\inf_{x \in \mathcal{X}}f(x,y,z) \leq f(x^*,y,z) \leq \sup_{x \in \mathcal{X}}f(x,y,z)$ holds. Proposition 1 only becomes nontrivial when $\mathcal{P}(\alpha)$ is constrained. However, since $\mathcal{P}(\alpha)$ is not constrained and can be any collection of distributions, Proposition 1 becomes trivial.

4. “To get an estimate of the upper bound, we need to identify the values of C that
render X′ ∪ C and Y ′ \ C most dissimilar. For a lower bound, we need to identify values of C that render X′ ∪ C and Y ′ \ C most similar.” — Why? I don’t understand the problem setting.

5. What is $\psi$ in Equation (3)? It seems undefined.

6. Why the optimization problem in Equation (3) is hard, but Equation (4) is solvable? Given that Equation (4) is also under the cardinality constraints, isn’t that NP hard?

**Q9 Complying With Reviewing Instructions:**

Yes

---

> ### Author Rebuttal · Authors · 2024-04-05
>
> We thank the reviewer for their comments. We are encouraged that they appreciate the significance of the problem we are tackling. Below we respond to the reviewer’s main questions
>
> 1. **“Grouping” of variables and the lack of clarifying examples**. We are sorry that the reviewer found this definition confusing. We agree with the reviewer that the best way to present our problem is through an example. In the introduction, we give a couple of examples. We also give a couple of examples in the experiment section. We restate one of these examples here for clarity: In settings where we wish to use the MMD to detect differences in genome sequences between healthy individuals and patients with myocardial infarction (MI). Ideally, we would have two matrices, one with the genome sequences of healthy patients and one with the genome sequences of MI patients. We can then directly apply the MMD to those two samples. Unfortunately, some MI cases typically go undiagnosed (more often than not, these are women’s cases, see the Merz reference in our paper). This means that in our observed data, some of the MI cases are incorrectly labeled as healthy. In this case, we say that some of the MI cases are incorrectly grouped with the healthy cases. We will clarify that example more in the writing, and we will highlight it more throughout the paper.
>
> 2. **Proposition 1.** We wish to clarify that proposition 1 is not just stating that
> $ \inf_{P_C \in \mathcal{P}(\alpha)}\text{MMD}(P_C,  P_{X'}, P_{Y'}) \leq \text{MMD}(P_{C^*},  P_{X'}, P_{Y'}) \leq \sup_{P_C \in \mathcal{P}(\alpha)}\text{MMD}(P_C,  P_{X'}, P_{Y'})$. Importantly, the proposition establishes that these are the _sharpest possible bounds_. In a sense, it is an impossibility result: it establishes that you cannot get better estimates that the ones we provide in the paper (+/- estimation errors, which are studied at length in the rest of the paper). We do agree with the reviewer that this result is simple and intuitive, which we believe adds strengths to our findings.
>
> 3. **The definition of $\psi$** is included in equation 3, where $\psi$ is introduced. We state it here for completeness
> $$\psi(C, X', Y')=\frac{(1 - \epsilon)}{n}\sum_i \sum_j k(x'_i, c_j)-\frac{(1+\epsilon)}{n} \sum_i \sum_j k(y'_i, c_j)+\frac{\epsilon}{n} \sum_i \sum\_{j \not= i} k(c_i, c_j)$$
> Intuitively, $\psi$ can be thought of as a weighted variant of the typical witness function, where the typical terms of the witness function are weighted by $(1 - \epsilon), (1 + \epsilon)$ and $\epsilon$ respectively. We will add this intuition in the final version of the paper.
>
> 4. **The reviewer is correct, $P_C$ is a distribution, not the RKHS**. As we mention in the paragraph below definition 1, we drop the RKHS space from the notation of the MMD since it is fixed to be the RKHS with bounded norm, simplifying the MMD notation to be MMD$(P_{X’}, P_{Y’})$. In addition, we state right before and after equation 2 that we introduce some abuse of notation to explicitly encode the dependence of the MMD on the contaminated set, describing MMD$(P_{X}, P_{Y})$ = MMD$(P_{C*}, P_{X’}, P_{Y’})$, where $P_{X}, P_{Y}$ are the uncontaminated distributions, and $P_{X’}, P_{Y’}$ are the contaminated distributions and $P_{C*}$ is the unknown distribution over the true contaminated set. This abuse of notation was necessary due to space limitations and we apologize for the confusion it caused. We will dedicate more space to clarifying our notation in the final version.
>
> 5. **Confusion about the problem setting**: To get an upper bound on the MMD, we need to find the data points that, if swapped, would maximize the MMD. To get a lower bound on the MMD, we need to get the data point that, if swapped, would minimize the MMD.
>
> 6. The reviewer is correct in that equation 3 and equation 4 give the same dimension of $\hat{C}$ (i.e., their solutions would have the same cardinality). However, equation 4 is “allowed” to pick any value of $\hat{C} \in R^d$ (where $d$ is the data dimension). By contrast, equation 3 has to pick values of $\hat{C} \in Y’$. We believe the current write up does not fully clarify this point, we will explicitly state the explanation given here in the main paper. We thank the reviewer for highlighting this.

---

### Official Review · Reviewer_vJ1C · 2024-03-24

**Q2-1 Originality-Novelty:** 3
**Q2-2 Correctness-Technical Quality:** 3
**Q2-5 Clarity Of Writing:** 3

**Q1 Summary And Contributions:**

The paper addresses the estimation of the Maximum Mean Discrepancy (MMD), a nonparametric measure of distance between two distributions, in the presence of $\epsilon$-contamination where a fraction $\epsilon$ of samples from one distribution is incorrectly mixed with the other. The authors show that the conventional MMD estimate is unreliable under $\epsilon$-contamination and propose a method for partial identification of the MMD. They derive sharp upper and lower bounds for the MMD and develop an estimation technique that provides tight bounds with a low false coverage rate. Empirical validation on three datasets demonstrates the superiority of their approach compared to alternative methods, indicating faster convergence and improved accuracy.

**Q2-3 Extent To Which Claims Are Supported By Evidence:**

3: Good: the main claims are supported by convincing evidence (in the form of adequate experimental evaluation, proofs, (pseudo-)code, references, assumptions).

**Q2-4 Reproducibility:**

3: Good: key resources (e.g. proofs, code, data) are available and key details (e.g. proofs, experimental setup) are sufficiently well-described for competent researchers to confidently reproduce the main results.

**Q3 Main Strengths:**

1. Relaxing the assumption of error-free samples and studying the estimation of MMD under $\epsilon$-contamination.
2. Introducing partial identification of the MMD and deriving sharp upper and lower bounds for the contaminated scenario.
3. Proposing a novel estimation method for these bounds, which outperforms alternative approaches in terms of convergence rate and accuracy.

**Q4 Main Weakness:**

While the assumption of a known value for $\epsilon$ is not overly sensitive, it can still be considered a strong assumption.

**Q5 Detailed Comments To The Authors:**

1. Could you please provide information on the computational complexity of the proposed method?

2. What factors contribute to the superior performance of the proposed method in high-dimensional, low-sample-size scenarios compared to QNO in Table 1?

3. What is the primary challenge in developing an adaptive estimation technique that does not rely on foreknowledge of $\epsilon$?

**Q9 Complying With Reviewing Instructions:**

Yes

---

> ### Author Rebuttal · Authors · 2024-04-05
>
> We thank the reviewer for their thoughtful comments. We are encouraged that the reviewer recognizes the importance of partial identification in the context of MMD estimation, and the novelty of our estimation approach. We respond to the reviewer’s questions below.
>
> 1. **Regarding the complexity.** The reviewer’s insightful question prompted us to do a comparison of wall clock time in seconds vs. $\epsilon.$ [This](https://anonymous.4open.science/r/mmd_wallclock_time_uai2024/sensitvity_time.jpg) anonymous link has the results of this comparison on the Forest dataset, with $n=2000$, showing our basic (non-stepwise) approach is faster than the QNO approach, and our stepwise approach is faster than the stepwise QNO variant. We will include this plot in the appendix. To directly answer the reviewer’s question: For each step in our algorithm, we need to compute the witness function, which essentially is the kernel matrix. This operation is usually in the order of $O(n^2d)$. We note that this step is necessary for our algorithm as well as others, and some of the calculated values can be cached for reuse in subsequent iterations of our algorithm. Finally, we take the quantile of the witness function, which is essentially a selection algorithm that can be solved in linear time (i.e., $O(n)$).
>
> 2. The reviewer raises an excellent question about **why our approach does better than QNO in the high dimensional and low sample size settings**. In general, we do better in low sample size settings because unlike QNO, our approach’s finite sample efficiency does not depend on the size of the contaminated set. This is the key finding from section 3. In the high dimension setting, as we mention in section 5.1, we do better because the QNO optimization objective depends on the data dimension. To expand on this explanation, let’s consider a simple example where there is only one contaminated sample, i.e., $|C| = 1$. For high dimensional settings, QNO solves the following optimization problem:
> $$\hat{c} = \frac{\partial \psi(c, X’, Y’)}{\partial c},$$ where $\psi(.)$ is as defined in equation 3 in the main text. The important detail here is that the parameter that QNO is solving for (that is $c$) has the same dimension as the observed data! By contrast, our approach reduces to estimating a quantile of the empirical witness function. This quantile is a scalar (since the empirical witness function values are one dimensional).
>
> 3. **Knowledge of $\epsilon$**. We chose to focus on a setting where we have some (at least rough) knowledge of $\epsilon$ because it matches many real data settings. However, we can still construct estimated bounds without such knowledge. For meaningful settings, we know that $0 < \epsilon < 1$. One possible approach here is to estimate our bounds for a sequence of $\epsilon_1 = 0.01, …, \epsilon_k = 0.5, …,  \epsilon_K = 0.99$, and compute the bounds $\overline{\mu}_k, \underline{\mu}_k$ for each level. The final bounds can be taken as $ \min_k \underline{\mu}_k, \max_k \overline{\mu}_k$.
> We believe this is a very interesting point, and we will add it to the final paper. We thank the reviewer for their insightful question.

---

### Meta-Review · Area_Chair_ES41 · 2024-04-11

This work studies estimation of the maximum mean discrepancy using $\epsilon$-contaminated data. All of the reviewers suggested that the problem and approach are interesting, even if a bit incremental. Several reviewers praised the clarity of the theoretical results and the novelty of the proposed estimator.

A final note: Reviewer uY2t was asked to modify their review by both the PCs and me and did not do so. Without this modification it is unclear whether that reviewer's recommendation is based on concerns about overlap with the work on arXiv - which it should not be - or the other issues they raised. Given this, I have decided not to take Reviewer uY2t's recommendation into account when writing this meta review.